# Nonlinear DNA methylation trajectories in aging male mice

Maja Olecka [1,5], Alena van Bömmel [1,5], Lena Best [2], Madlen Haase[3], Silke Foerste[1], Konstantin Riege[1], Thomas Dost [2], Stefano Flor [2], Otto W. Witte[3], Sören Franzenburg [4], Marco Groth [1], Björn von Eyss [1], Christoph Kaleta [2,6], Christiane Frahm[3,6] & Steve Hoffmann [1,6] ✉

Although DNA methylation data yields highly accurate age predictors, little is known about the dynamics of this quintessential epigenomic biomarker during lifespan. To narrow the gap, we investigate the methylation trajectories of male mouse colon at five different time points of aging. Our study indicates the existence of sudden hypermethylation events at specific stages of life. Precisely, we identify two epigenomic switches during early-to-midlife (3-9 months) and mid-to-late-life (15-24 months) transitions, separating the rodents' life into three stages. These nonlinear methylation dynamics predominantly affect genes associated with the nervous system and enrich in bivalently marked chromatin regions. Based on groups of nonlinearly modified loci, we construct a clock-like classifier STageR (STage of aging estimatoR) that accurately predicts murine epigenetic stage. We demonstrate the universality of our clock in an independent mouse cohort and with publicly available datasets.

Biological aging, which we refer to as a process taking place over the whole lifespan, is frequently perceived as a constant decay of function at the cellular, tissue, and organismal level, e.g., caused by the accumulation of DNA damage, telomere shortening, loss of proteostasis or stem cell exhaustion[1]. Thus, many studies focus on discovering linear relationships between time and various molecular data. Since the aging process involves abrupt changes as well, e.g., menopause or loss of neurons in the intestine[2], linear models may be too simplistic, and a more thorough search for nonlinear trajectories may shed new light on the nature of aging and its driving forces.

In recent years, the scientific literature has increasingly appreciated nonlinearities in the aging process. For example, studies in *Drosophila* suggest discrete stages in aging, marked by a sudden increase in intestinal permeability[3]. A study on aging human plasma proteome found that most changes across the lifespan are nonlinear and occur in waves around 34, 60, and 78 years[4]. Another study of human transcriptomes found an acceleration of cancer incidence in the mid-life phase between 35–45 years, potentially pointing to an acceleration of DNA damage[5]. In addition, Schaum et al. characterized nonlinearities in gene expression levels in aging mouse organs[6], and Kang et al. described nonlinear aging patterns in muscle transcriptomes of old mice[7]. Another study on aging skin identified four distinct aging phases based on integrated epigenetic and transcriptomic features[8].

However, research on discontinuous aspects of aging is still in its infancy, and critical molecular processes, including those involving epigenomic regulation, have yet to be examined.

DNA methylation (5mC) is an essential layer of epigenetic control. Although DNA methylation studies have revealed specific age-related phenomena, such as global demethylation and hypermethylation of

[1]Hoffmann Lab, Leibniz Institute on Aging - Fritz Lipmann Institute (FLI), Beutenbergstrasse 11, 07745 Jena, Germany. [2]Research Group Medical Systems Biology, Institute for Experimental Medicine, University of Kiel and University Medical Center Schleswig-Holstein, 24105 Kiel, Germany. [3]Department of Neurology, Jena University Hospital, Am Klinikum 1, 07747 Jena, Germany. [4]Institute of Clinical Molecular Biology, Kiel University and University Medical Center Schleswig-Holstein, 24105 Kiel, Germany. [5]These authors contributed equally: Maja Olecka, Alena van Bömmel. [6]These authors jointly supervised this work: Christoph Kaleta, Christiane Frahm, Steve Hoffmann. ✉e-mail: Steve.Hoffmann@leibniz-fli.de

CpG islands (reviewed in ref. 9), we still know little about the dynamics of this epigenetic mark. Notably, today's methylation-based epigenetic clocks are frequently built using linear regression models (usually involving elastic-net regularization) and thus geared to detect continuous changes (reviewed in ref. 10). So far, attempts to distill nonlinear age-related changes, e.g., using the power-law model[11], have not yielded a comprehensive picture.

In the context of aging, the intestine is of particular interest. While the tissue has one of the highest regenerative capacities, numerous biological functions undergo age-related alterations, affecting intestinal microbiota, barrier, immune functions, and the enteric nervous system (reviewed in ref. 12). At the same time, the consequences of such changes can have a profound systemic impact on the emergence of age-related phenotypes in other organs, such as the brain, the heart, or the endocrine system. In addition, Dambroise et al. suggest that disruption of the intestinal barrier may be a critical evolutionary conserved event in the aging process[13].

In this work, we perform genome-wide DNA methylation profiling to identify and characterize 5mC trajectories in the aging male mouse colon. We demonstrate the existence of nonlinear methylation trajectories in aging at specific sets of CpGs associated with developmental processes and the nervous system. Moreover, we show that nonlinearly modified CpGs summarized by centroids provide a robust set of predictor variables to estimate the epigenetic life stages. Our analysis strategy is shown in Fig. 1a.

## Results

To analyze DNA methylation patterns in aging, we performed reduced representation bisulfite sequencing (RRBS) on colon DNA samples from 83 male C57BL/6J/Ukj mice at five time points throughout life (3, 9, 15, 24, and 28 months). Utilizing the MspI restriction enzyme[14], RRBS enriches GC-rich genomic regions often associated with critical regulatory elements such as promoters and enhancers. Principal component analysis (PCA) of raw methylation data reveals that the first principal component, representing more than 25% of the data's total variance, clearly covaries with the age of the samples (Fig. 1b and Supplementary Fig. 1a). Unsupervised hierarchical clustering carried out on the same data suggests a separation of the samples into three distinct primary life stages, i.e., early life (3 mo), midlife (9 mo–15 mo), and late-life (24 mo–28 mo) (Fig. 1c).

### Differentially methylated regions in aging mouse intestine

To focus on the greatest age-related methylation changes, we restricted our analysis to cytosines located in differentially methylated regions during aging (aDMRs). In particular, we calculated pairwise aDMRs between all combinations of time points. Combining all pairwise results yielded 18,006 aDMRs comprising 84,693 CpGs (5.5% of the whole dataset) (Supplementary Data 1). As expected, the highest number of aDMRs was detected between 3 mo and 28 mo ($n = 4262$). The smallest number was seen while comparing 24 mo and 28 mo mice ($n = 38$; Fig. 1e). Notably, we observed a substantial overlap between aDMRs and reduced their number to 3981 when merging the overlapping regions. Pairwise similarity based on the number of aDMRs supports the aggregation of the samples into three distinct aging stages– young, midlife, and late-life, which was also found in the unsupervised hierarchical clustering of methylation levels (Fig. 1c). On a global level, we observe gradual hypermethylation of aDMRs during aging. Notably, the majority of aDMR-associated CpGs in 3 mo animals is unmethylated (Fig. 1d). Furthermore, aDMRs are significantly enriched in exons, and CpG islands as compared to randomly drawn regions from the RRBS background (Supplementary Fig. 1b), and 91% of aDMR-associated cytosines could be annotated to a gene or a promoter. Surprisingly, aDMRs are significantly enriched in genes with specific functions for the cerebral cortex (Fig. 1f) as well as in canonical pathways and potential upstream regulators associated with the

nervous system (Fig. 1g). The most enriched biological processes with aDMRs are related to DNA pattern recognition, synapse organization, and forebrain development and the most enriched molecular functions are DNA binding and channel activity, both pointing to transcription factor activity and processes in the nervous system (Supplementary Fig. 1e). We speculated that the aDMRs might be linked to age-related changes in the enteric nervous system and found a significant enrichment in markers of enteric neurons and glial cells compiled by ref. 15 (OR = 4.2, $p$val = 5.3e-03 and OR = 4.1, $p$val = 3.2e-03, respectively, Fisher's exact test). For instance, we found age-related DNA methylation changes in the first exon of *Kctd8* (Fig. 1h), encoding a subunit of the GABA-B receptor, which is broadly expressed in the nervous system and plays an essential role in controlling neuronal excitability. Further, we found age-related methylation changes in other markers of enteric neurons (*Elavl4*, *Scg3*, *Crmp1*, and *Gdap1l1*, Supplementary Fig. 1c) and glial cells (*Itga4*, *Pdpn*, *Ptprz1*, and *Sox10*, Supplementary Fig. 1d).

The enteric nervous system is thought to be more vulnerable to degeneration and cell death during aging than other parts of the nervous system (reviewed in ref. 16), and these results may indicate that epigenetic changes contribute to this vulnerability.

### Nonlinear DNA methylation trajectories in aging

To identify recurrent trajectories of cytosine methylation, we performed unsupervised clustering of aDMR-associated CpGs using *clust*[17]. After four iterative rounds of clustering, we obtained 19 clusters comprising 94% of all aDMR-associated cytosines (Supplementary Data 2). Analyzing the methylation Z-scores in the clusters, we identified two clusters with linear methylation trajectories (C1 and C5), while all remaining clusters exhibited nonlinear methylation changes (Fig. 2a). Consequently, almost half ($n = 34,322$) of all clustered aDMR cytosines ($n = 71,447$) were assigned to clusters with nonlinear methylation trajectories.

We selected the five most prominent clusters representative of different trajectories for further analysis. This selection includes the largest linear cluster, C1, representing global age-associated hypermethylation. Similarly, we find another yet smaller linear cluster, C5, exhibiting hypomethylation during aging (Fig. 2a). Additionally, we identified three clusters showing a markedly nonlinear behavior during specific life stages (Fig. 2a). Specifically, clusters C2 and C3 show abrupt methylation increases during the early-to-mid-life transition between month 3 and month 9. One of the genes potentially affected by CpGs following this trajectory is *Zcchc3* (Fig. 2d), zinc finger CCHC-type containing three genes which play a vital role in the innate immune system[18,19]. Other genes with the most CpGs associated with clusters C2 and C3 are *Ajap1* and Protocadherins Gamma genes (*Pcdhg*, Supplementary Fig. 2a). Ajap1 is a structural protein involved in numerous human malignancies and correlates with tumor growth and survival[20]. The cell surface glycoproteins Protocadherins Gamma (Pcdhga1-9, Pcdhgb1-4) are linked to differentiation, cancer, aging, neurological disorders, and muscle weakness[21,22].

Analogously, cluster C4 exhibits sudden hypermethylation during the mid-to-late-life transition (>15 mo), as shown for gene *Nkx6-2* (Fig. 2d). NK6 Homeobox 2 transcription factor regulates multiple developmental processes with a leading role in neurogenesis[23,24] and pancreatic development[25]. Interestingly, this gene was recently found to be associated with delta age, a biomarker of brain aging that captures differences between the chronological age and the predicted biological brain age[26]. Moreover, cluster C4 is strongly linked to many other genes with known age-associated functions, such as *Satb1*, *Nova1*, *Adra2c*, *Pax5*, *Zeb1*, and *Dmrta2*[27–35] (Supplementary Fig. 2a).

To explore the epigenomic context of cytosines from different clusters, we evaluated the enrichment of ENCODE chromatin states of murine intestines[36] at the time of birth (P0) in terms of odds ratios (OR) (Fig. 2b). Notably, cytosines associated with different life stage transitions show apparent differences in their epigenomic context. Also,

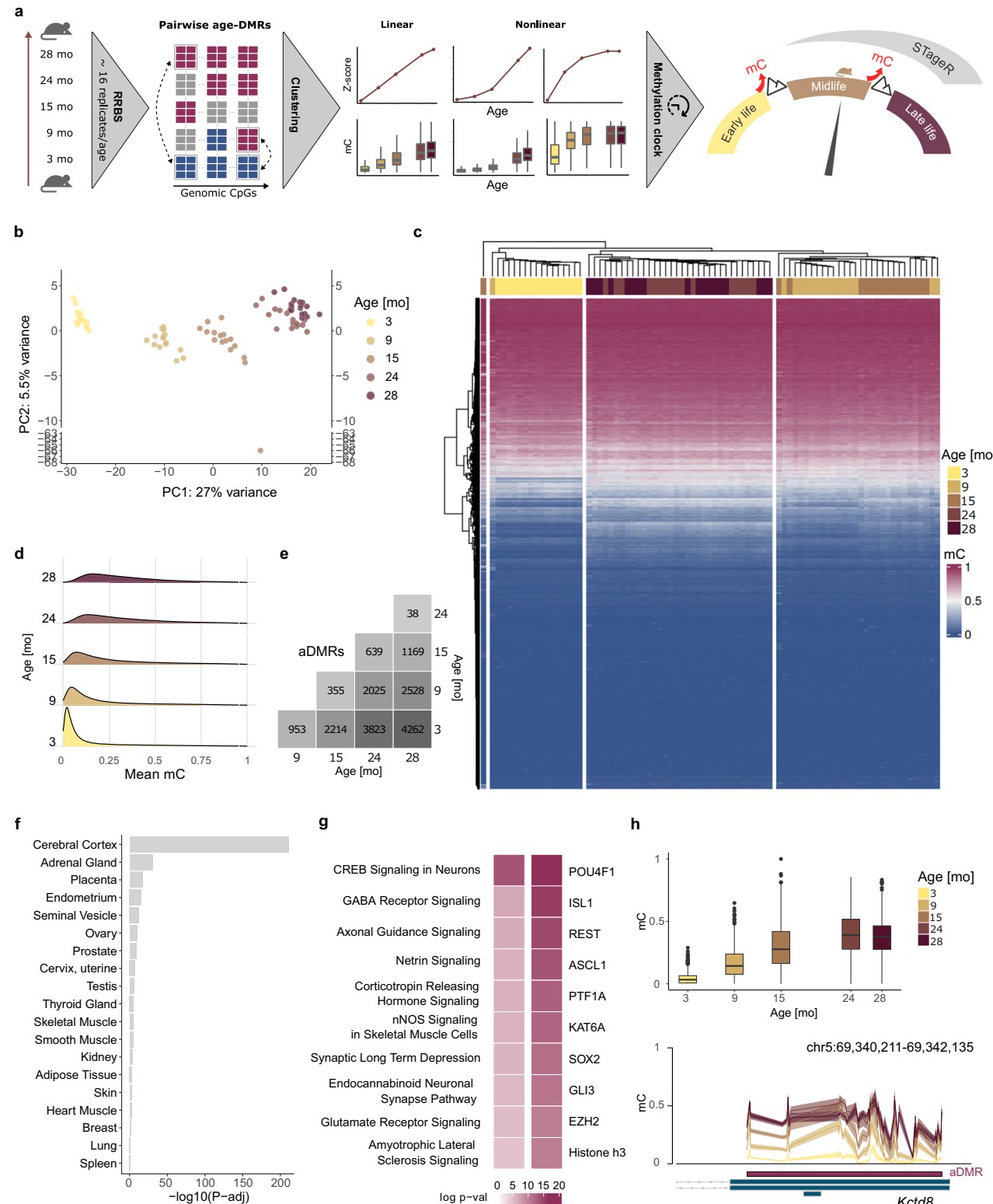

clusters with similar methylation trajectories show similar epigenetic patterns and are grouped together by hierarchical clustering (e.g., C3, C17, C13, and C2 or C4 and C19, Fig. 2b). Cytosines affected by mid-to-late-life transition in C4 are typically lowly methylated, show comparably minor methylation gains (Fig. 2a), and are strongly enriched in bivalent promoters (OR = 25). Bivalent chromatin is characterized by simultaneous occupancy with activating (H3K4me3) and repressing (H3K27me3) histone marks and is often involved in regulating developmental genes in stem cells[37–39]. Moreover, C4 CpGs are strongly

overrepresented at binding sites of Polycomb repressive complex 2 (PRC2) proteins (Suz12, EZH2, and Jarid2, Supplementary Fig. 3c), which repress gene expression at bivalent promoters and are implicated in cancers and developmental disorders (reviewed in ref. 40). Strikingly, 86% (4372 out of 5091) cytosines of C4 are found within binding sites of Mtf2, a recruiter of the PRC2 (Supplementary Fig. 3c).

In contrast, cytosines in C2 and C3 showing early-to-midlife transition accumulate in potent proximal enhancers and H3K9me3-associated heterochromatin (Fig. 2b). H3K9me3 is vital for

**Fig. 1 | DNA methylation dynamics in aging male mouse colon. a** Experimental strategy. Reduced representation bisulfite sequencing (RRBS) was performed on a set of colon DNA samples from 83 adult male mice at five ages. Clustering based on the Z-score of methylation levels of CpGs from pairwise differentially methylated regions revealed various methylation trajectories during aging. Most abundant nonlinear clusters divided lifespan into three stages and were used to construct an epigenetic clockwise classifier– STageR. Hypermethylation events separating the epigenetic stages of life are presented as mountain peaks. **b** Principal component analysis (PCA) based on 10,000 randomly selected CpGs. One sample has been identified as an outlier and removed from the dataset. PCA of all samples is presented in Supplementary Fig. 1a. **c** DNA methylation patterns in 10,000 randomly selected CpGs with hierarchical clustering of all samples indicate three major epigenetic life stages (early-, mid-, and late-life). **d** Methylation distribution at aDMR CpGs in the age groups; $n = 16$ (3 mo, 9 mo, 15 mo, 24 mo), $n = 18$ (28 mo) animals.

**e** Number of differentially methylated regions in aging (aDMRs) for all pairwise combinations of age groups. **f** Enrichment of tissue-specific genes in genes associated with aDMRs. FDR-adjusted hypergeometric one-sided $P$ values are shown. **g** Top ten enriched ingenuity pathways (left) and potential upstream regulators (right) of genes associated with aDMRs. FDR-adjusted one-sided $P$ values of Fisher's exact test are shown. **h** Methylation pattern in aDMR in the first exon of *Kctd8*. DNA methylation levels are shown as box plots for the age groups (top) and as mean levels (line) with confidence intervals (shaded area) in the genome browser (bottom); $n = 874$ (3 mo), $n = 834$ (9 mo), $n = 823$ (15 mo), $n = 841$ (24 mo), $n = 945$ (28 mo) according to available measurements of 55 CpGs in the corresponding age groups. For the box plots, the center line shows the median, the box limits show the first and third quartiles, and the upper and lower whiskers extend from the hinge to the largest or the lowest value no further than 1.5× the interquartile range (IQR) from the hinge. Source data are provided as a Source Data file.

heterochromatin organization and genome integrity[41] and for establishing and maintaining cell identity by repressing lineage-inappropriate genes (reviewed in ref. [42]). Moreover, it was shown that H3K9me3 recruits DNA methyltransferase 1, promoting the maintenance of DNA methylation[43]. Additionally, we observe a strong enrichment of cytosines from C2 and C3 in binding sites of the immunoregulatory transcription factor Gata1 (Supplementary Fig. 3c).

The cytosines from clusters C1–C4 follow a similar genomic distribution, with most cytosines located in introns, exons, and promoters (Supplementary Fig. 3d).

It is important to note that the only cluster with cytosines showing a gradual decrease of methylation during aging (C5; 1.2% of all clustered cytosines) differs markedly from its hypermethylated counterparts in terms of the epigenomic contexts (Fig. 2b) and genomic distribution with almost 80% of cytosines located in exons and introns (Supplementary Fig. 3d).

To further explore the potential function of genes associated with nonlinear trajectories, we have tested whether the clusters are enriched for gene sets associated with hallmarks of aging proposed by ref. [44]. To this end, we used lists of genes associated with the nine aging-related processes compiled by ref. [8]. The odds ratios for the nine gene sets show a distinct pattern for the prominent clusters C1–C5 (Fig. 2c). Interestingly, the mid-to-late-life cluster C4 is highly enriched for stem cell exhaustion which goes in line with the occupancy of bivalent chromatin by C4 cytosines. Moreover, C4 genes are associated with cellular senescence. The early-to-midlife clusters C2 and C3 are enriched for stem cell exhaustion and epigenetic alterations (Fig. 2c).

Next, we sought to evaluate whether DNA methylation at cytosines following nonlinear trajectories corresponds rigidly to chronological age or can be modified. To this end, we analyzed publicly available bisulfite sequencing datasets and found that 5mC levels respond to different experimental conditions. Specifically, mice with an intestinal infection[45] show older methylation profiles in nonlinear clusters compared to controls (Fig. 3). Strikingly, colon organoids derived from 40 days old mice[46] show an extremely aged methylation profile with almost full hypermethylation even after only 2 months of culture (Fig. 3).

In summary, we identified loci undergoing sudden hypermethylation during aging. Their nonlinear trajectories divide lifespan into three major epigenetic stages (Fig. 2d). Importantly, DNA methylation transitions are not accompanied by considerable shifts in cell type composition (Supplementary Fig. 4). Our data suggest that cytosines undergoing mid-to-late-life transition are associated with effects on PRC-controlled developmental programs, which may affect stem cell function. In contrast, changes between early life and midlife may reflect the maturation processes of the organ. Although cytosines from the two transitions differ, as well as the majority of associated genes (Supplementary Fig. 2b), gene ontology (GO) analysis indicated that affected genes are involved in similar processes, such as development,

regulation of transcription, and transsynaptic signaling (Fig. 2e and Supplementary Fig. 2c).

### Nonlinear gene expression trajectories in aging

To evaluate gene expression patterns of genes overlapping with CpGs following nonlinear DNA methylation trajectories, we conducted bulk RNA-Seq analysis on all the samples. The vast majority of genes from the nonlinear DNA methylation clusters are active (see Methods, Supplementary Fig. 3b). When comparing the gene expression deciles, we observe that genes associated with early-to-mid-life methylation changes show higher expression levels compared to those related to mid-to-late-life transition (Supplementary Fig. 3b). Interestingly, CpGs undergoing a nonlinear mid-to-late-life transition (C4) also enrich in genes differentially expressed between 15 and 24 months ($p$ value = 0.007, Fisher's exact test). Specifically, 13% of genes with C4 CpGs are also differentially expressed between 15 and 24 months.

To identify genes with the same trajectory on DNA methylation and expression levels, we clustered genes showing age-dependent dynamics based on their expression trajectory during the lifespan. To this end, we applied the same strategy as in the case of DNA methylation analysis. First, we identified all pairwise differentially expressed genes across the age groups ($n = 14,061$) and conducted five rounds of unsupervised clustering based on gene expression Z-scores. Interestingly, the largest gene expression cluster, CE1 ($n = 2483$), exhibits an expression gain between 15 and 24 months (Fig. 4a). Together with CE6 ($n = 801$), CE1 indicates the existence of mid-to-late-life transition on the expression level. Similar to the above, C4 CpGs significantly enrich in CE1 and CE6 genes ($p$ value = 0.02, Fisher's exact test). Additionally, we found cluster CE5 ($n = 761$) to show an early-to-mid-life transition.

Next, we identified 27 genes showing a simultaneous change in DNA methylation and gene expression during the early-to-mid-life transition and 146 genes during the mid-to-late-life transition (Fig. 4b, c). Many of the identified genes play essential roles in colon function. For instance, *Filip1l*, which undergoes an early-to-mid-life transition, acts as a tumor suppressor in mucinous colon cancer[47]. Additionally, 66 genes following mid-to-late-life trajectory show strong interactions on the protein level based on the STRING database (Fig. 4d). Strikingly, almost all genes from the network have known functions in colorectal cancer, intestinal barrier, or enteric nervous system. The most exciting examples include *Reln* and *Ntn*, which are associated with all these mentioned functions[48–54].

Overall, we identified genes that follow the same nonlinear trajectories on the gene expression level and DNA methylation of overlapping CpGs. These genes, particularly those associated with mid-to-late-life transition, encode crucial colon function and cancerogenesis regulators.

### Validation dataset

To validate the existence of nonlinear methylation trajectories during aging, we analyzed an independent set of 20 male C57BL6/J mice at

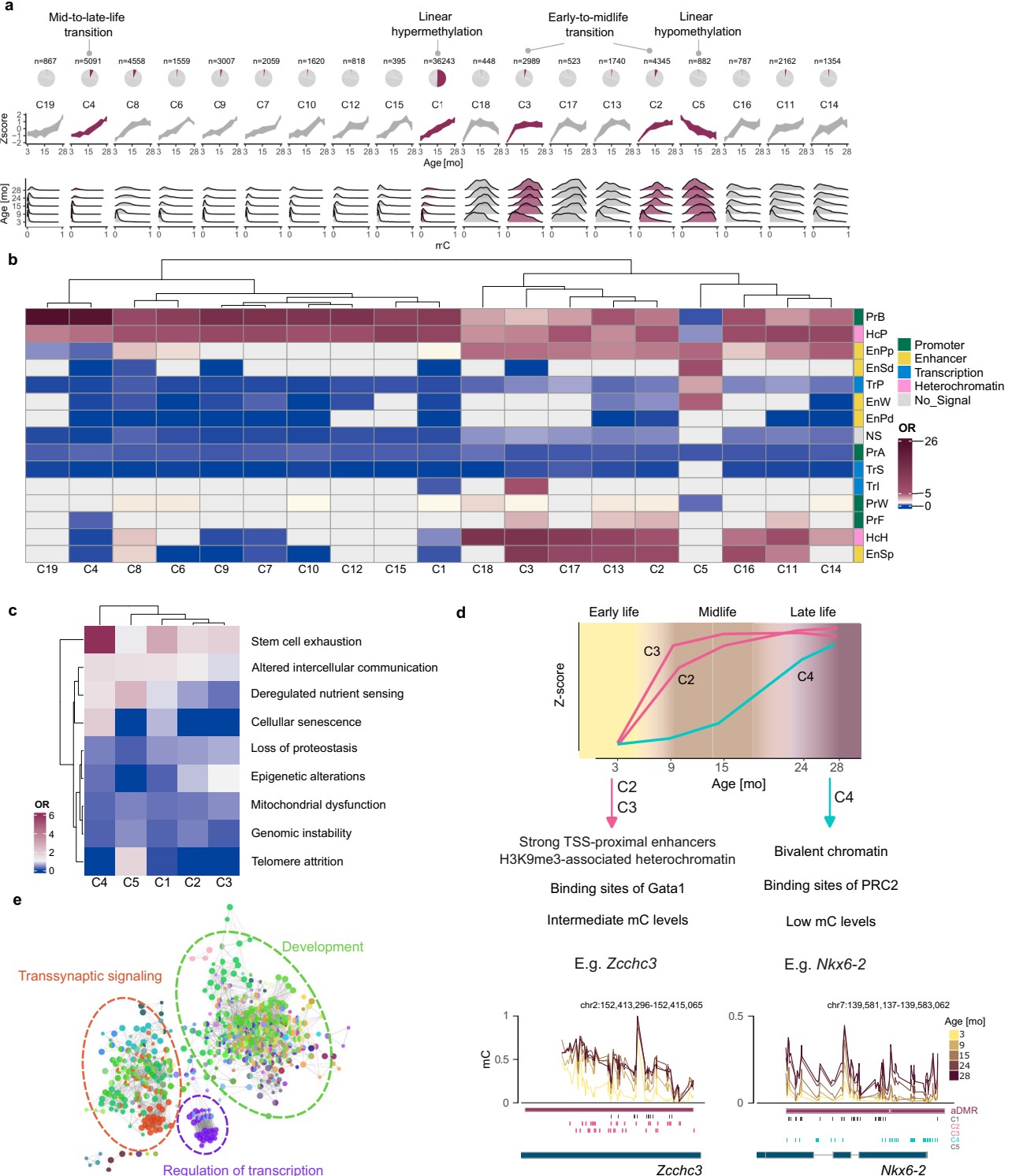

**Fig. 2 | All 19 methylation trajectories identified in the study. a** From top to bottom: pie charts with the cytosine proportion in the particular cluster to all aDMR cytosines; DNA methylation trajectories during aging as Z-scores and methylation distribution per age group. Red shading marks clusters chosen for further analyses. **b** Enrichment of chromatin segments obtained from ChromHMM in each cluster. PrA - Promoter, Active, PrW - Promoter, Weak, PrB - Promoter, Bivalent, PrF - Promoter, Flanking Region, EnSd - Enhancer, Strong TSS-distal, EnSp - Enhancer, Strong TSS-proximal, EnW - Enhancer, Weak, EnPd - Enhancer, Poised TSS-distal, EnPp - Enhancer, Poised TSS-proximal, TrS - Transcription, Strong, TrP - Transcription, Permissive, TrI - Transcription, Initiation, HcP - Heterochromatin, Polycomb-associated, HcH - Heterochromatin, H3K9me3-associated, Ns - No significant signal. **c** Enrichment of hallmarks of aging gene sets in genes associated with five representative clusters. **d** A scheme presenting nonlinear DNA methylation trajectories dividing lifespan into three stages and their main features. **e** Gene ontology (GO) analysis of genes associated with clusters C2, C3, and C4. GO terms were functionally grouped. A comparative analysis of C2 and C3 vs C4 genes is presented in Supplementary Fig. 2c. In **b**, **c** OR odds ratio. Source data are provided as a Source Data file.

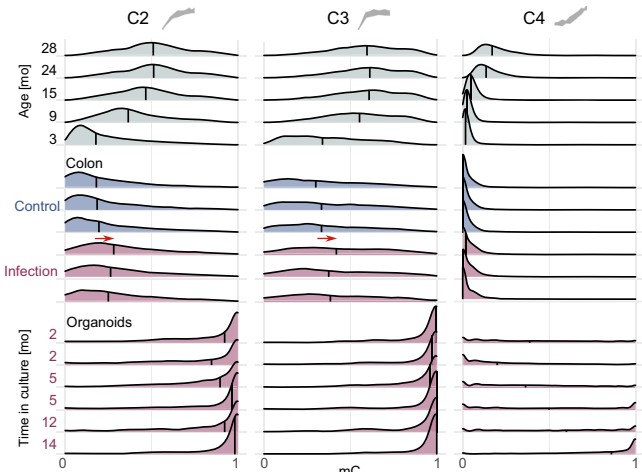

**Fig. 3 | DNA methylation of CpGs following nonlinear trajectories in publicly available bisulfite sequencing datasets.** Upper panel, DNA methylation of CpGs from most abundant nonlinear clusters C2, C3, and C4 from this study. Middle panel, reduced representation bisulfite sequencing (RRBS) data of colon samples from 27 weeks old mice with and without infection with *Helicobacter hepaticus*[45]. Red arrows highlight the shifts in the median between the two experimental conditions. Bottom panel, RRBS of colon-derived organoids[46]. The culture was established from the colon crypts of 40-day-old mice and continued for the periods indicated by the labels. Vertical lines in the density plots correspond to the methylation median. Source data are provided as a Source Data file.

four ages from another mouse facility: 3 mo ($n = 4$), 7 mo ($n = 5$), 12 mo ($n = 5$), and 27 mo ($n = 6$). To exclude potential artifacts, animal and sequencing facilities differed in the validation experiment. We used the same methodology to identify aDMRs in the validation set, resulting in 3336 merged aDMRs comprising 103,151 CpGs. In total, 74% (62,782) of the aDMR CpGs from our original analysis are located in aDMRs identified in the validation dataset—a surprisingly high value given all technical and biological variabilities (Fig. 5a). Likewise, the majority of the pairwise aDMRs determined in the original dataset could be validated by the independent set when the most similar age groups were considered (Supplementary Fig. 6a).

Further, we evaluated age-dependent methylation trajectories in the validation dataset. The validation data confirmed the trends in the linear clusters C1 and C5 and the nonlinear clusters describing early-to-midlife (C2 and C3) and mid-to-late-life transitions (Fig. 5b–f). Likewise, the methylation distribution in the young (3 mo), midlife (7–12 mo), and late-life mice (27 mo) is consistent with the methylation distribution in the original dataset in the corresponding life stage. Accordingly, the methylation Z-scores derived from the validation data follow the same linear trajectories in C1 and C5 (Fig. 5b–f) and nonlinear trajectories in clusters C2–C4 (Fig. 5c–e).

Taken together, our validation experiment confirms the linear and nonlinear methylation dynamics in aging mouse intestines.

**Cluster-based epigenetic clock STageR**
Epigenetic clocks modeling the chronological age from 5mC levels are typically built using supervised machine learning methods such as elastic-net regression (reviewed in ref. 10). The application of these models typically relies on a fixed set of CpGs, i.e., the model CpGs. One of the shortcomings of such clocks is their lack of transferability, especially when using the popular RRBS protocol, where the number of sites captured and sufficiently covered is highly variable between datasets. To remedy this shortcoming, we propose to build aging clocks based on measures derived from highly correlated and thus redundant CpG methylation information. For instance, using the methylation centroids of the above-described clusters instead of single CpG values.

Here, we propose an epigenetic clockwise classifier STageR (STage of aging estimatoR) which predicts the underlying aging-stage based on limited methylation information. In particular, we perform a multinomial logistic elastic net regression to predict the aging-stage membership (early-, mid-, or late-life) of each sample based on the median methylation level of CpGs in three nonlinear clusters, C2, C3, and C4. Thus, in our case, the dimensionality of the feature space is dramatically reduced from more than 80,000 age-associated CpGs to only three methylation clusters (see the training set in Supplementary Fig. 7a). Our choice of early-to-mid-life (3–9 mo) and mid-to-late-life (15–24 mo) clusters is underpinned by a number of important observations: (i) hierarchical clustering of raw methylation data already results in the division of samples into three main groups, i.e., 3 mo, 9 and 15 mo, 24 and 28 mo (Fig. 1c), (ii) early-to-mid-life (C2 and C3), as well as mid-to-late-life (C4) methylation clusters, together with C1 and C6, were detected during the first clustering round with clust[17], which indicates that the CpGs in these clusters exhibit a comparably strong signal, (iii) gene expression trajectories (Fig. 4a) support the timing of transitions identified on the level of DNA methylation. Specifically, cluster CE1 (the largest cluster) and CE6 support mid-to-late-life transition, and CE5 supports early-to-mid-life transition.

In a tenfold cross-validation, our methylation clock predicts the aging-stage membership without errors. Interestingly, the model achieves this optimal performance by reflecting the stage transitions in the model coefficients (Fig. 6b). For example, the early-to-midlife cluster C3 with the steep increase between 3 mo and 9 mo, has the largest positive coefficient in the early life stage. The other early-to-midlife cluster, C2, has the highest coefficient in the midlife stage. Differently, the mid-to-late-life cluster C4 has the largest positive coefficient in the late-life stage (Fig. 6b).

To test the robustness of STageR, we randomly sampled a set of cytosines of a given size from each cluster, calculated then the cluster medians and used them for testing in each fold of cross-validation. Subsequently, we evaluated the misclassification error using a tenfold cross-validation. The entire procedure was repeated ten times, choosing different cytosines randomly in each repetition. Strikingly, the median misclassification error remained smaller or equal to 10% for a minimum of 100 cytosines per cluster (Fig. 6c). It increased to 13% for as little as 30 cytosines per cluster and rose over 20% only when ten or fewer cytosines per cluster were used.

Next, we tested STageR on the validation dataset. In summary, all 20 validation samples were correctly assigned to their aging stages, where the 3 mo mice were classified in early life, 7 mo and 12 mo mice in mid-life, and 27 mo mice in the late-life stage (Fig. 6d). Interestingly, the predicted probabilities of all three stages reflect the younger age of all 7 mo samples in comparison to 12 mo samples with larger predicted probabilities of early life stage while having their maximum in the midlife stage (Fig. 6d).

Subsequently, we tested STageR's performance with incomplete methylation information. We sampled a fraction of overlapping cytosines in the validation dataset (75, 50, 25, and 10% of the original size, respectively) in each cluster, then calculated the sample median methylation and predicted the aging-stage using STageR. The results based on sampled medians confirm the robustness of STageR even for a small number of cytosines (Fig. 6e). The midlife (12 mo) and late-life (27 mo) samples were consistently correctly classified even when medians were determined on only 10% of the cluster cytosines. Early life samples were correctly classified using at least 25% of cytosines, showing a low misclassification rate (0.12) when using 10% of cytosines. A larger variability involving misclassifications was found only for the 7 mo samples when medians were based on 25% or less of all cytosines when these samples were partially misclassified to the early life stage. However, the mean predicted probabilities show consistent

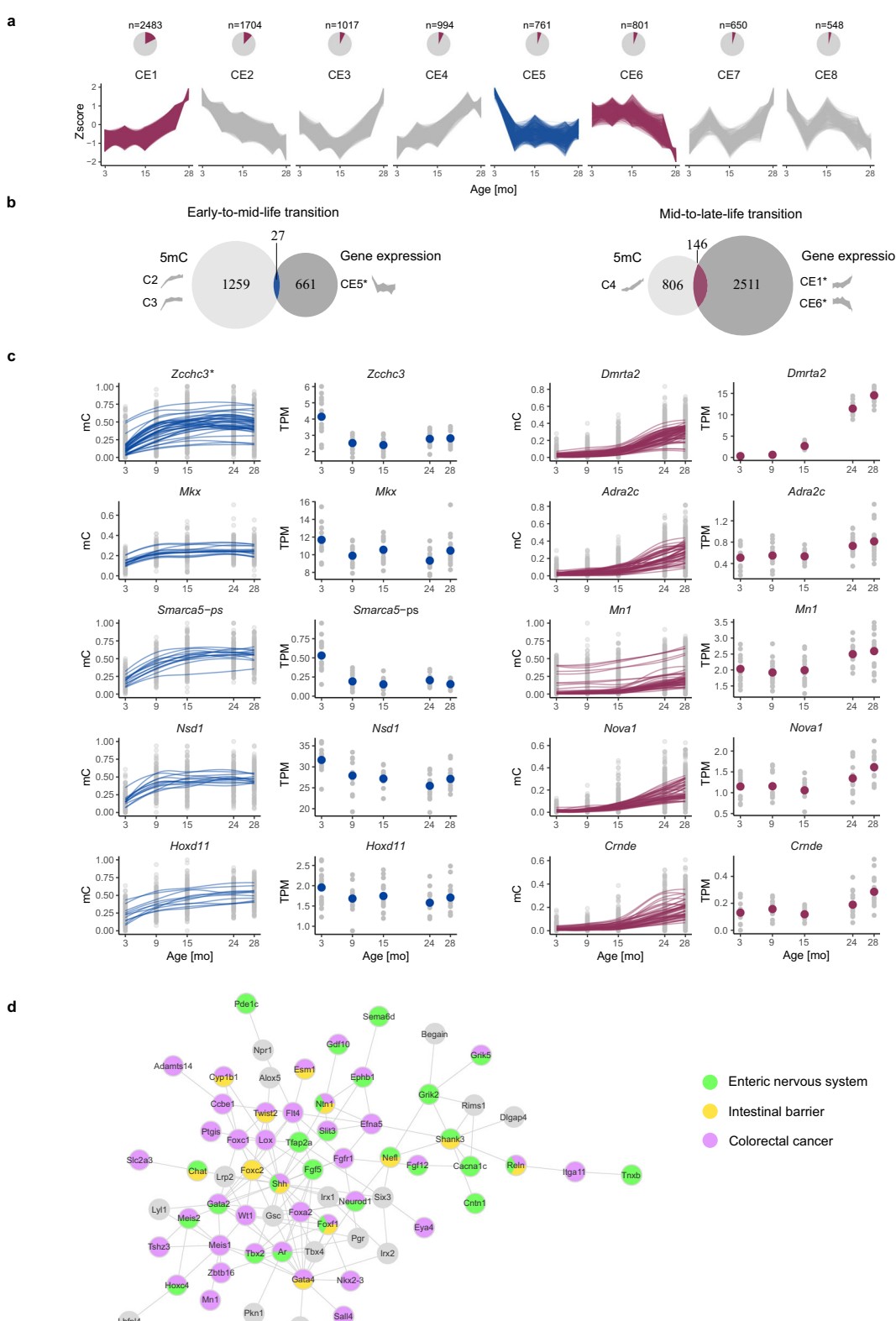

patterns for all subsampled samples and their corresponding aging stages (Fig. 6e).

Finally, we tested STageR on publicly available datasets[45,46]. The proportion of cytosines from clusters C2, C3, and C4 covered by the analyzed samples ranged from 20 to 80% (Supplementary Fig. 7b). We were able to correctly predict the aging-stage of 27 weeks (6,75 months) old mice based on their methylation levels measured in colon[45] (Fig. 5f). Interestingly, STageR confirmed our previous observation of older methylation profiles in mice with an intestinal infection compared to controls. Specifically, STageR assigns higher probabilities of the midlife stage for mice infected with *Helicobacter hepaticus* than healthy mice (Fig. 6f). Strikingly, colon organoids derived from 40 days old mice[46] are classified into late-life stages independently from time in culture (Fig. 6f).

Taken together, we defined a cluster-based epigenetic stage clock STageR that can capture the aging-related methylation dynamics even

**Fig. 4 | Nonlinear gene expression trajectories in aging mouse colon. a** Pie charts illustrating the gene proportions within specific clusters relative to all differentially expressed genes (top). Gene expression trajectories during aging as Z-scores (bottom). Blue shading—cluster corresponding to early-to-mid-life transition, red shading—clusters corresponding to mid-to-late-life transition. **b** Genes overlapping with at least one CpG following early-to-mid-life or mid-to-late-life methylation trajectory were intersected with genes following the same type of transition on gene expression level. * - only genes covered in the RRBS dataset were included in the analysis. **c** Top five genes with the highest number of CpGs undergoing either early-to-mid-life transition (left) or mid-to-late-life transition

(right) were selected from the intersection sets depicted in **b**. Each smoothed line corresponds to one CpG. Colored dots represent the mean TPM for an age group. * - the same set of CpGs was annotated to *6820408C15Rik* (Supplementary Fig. 5) **d** STRING protein–protein interaction network created from genes following a mid-to-late-life trajectory on both DNA methylation and gene expression levels. Genes with known functions in the enteric nervous system, intestinal barrier, and/or colorectal cancer are marked with specific coloring. Supplementary Table 1 contains corresponding references related to the functions of genes from the network. Source data are provided as a Source Data file.

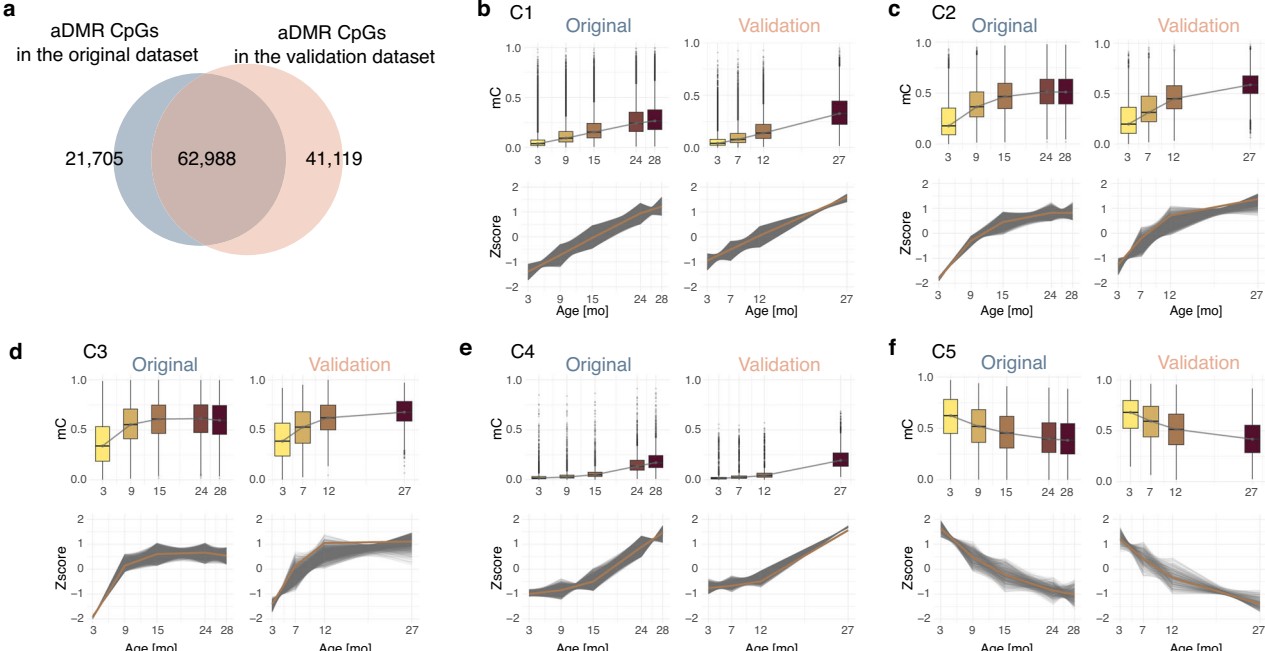

**Fig. 5 | Validation dataset. a** Overlap between differentially methylated regions in aging (aDMR) CpGs in the original and validation datasets. **b–f** Distribution of mean methylation levels by age group in the original dataset (left) and in the validation dataset (right) and the methylation Z-score trajectories during aging for clusters C1–C5, respectively. The yellow line indicates the cubic smooth spline of the Z-scores in the original dataset. For the validation dataset, 70% of the cytosines closest to the original trajectory in each cluster are plotted. Number of CpGs:

$n = 36,243$ (C1 original) $n = 24,920$ (C1 validation); $n = 4345$ (C2 original) $n = 2956$ (C2 validation); $n = 2989$ (C3 original) $n = 2025$ (C3 validation); $n = 5091$ (C4 original) $n = 3502$ (C4 validation); $n = 882$ (C5 original) $n = 602$ (C5 validation). For the box plots, the center line shows the median, the box limits show the first and third quartiles, and the upper and lower whiskers extend from the hinge to the largest or the lowest value no further than 1.5× the interquartile range (IQR) from the hinge. Source data are provided as a Source Data file.

when using a minor proportion of the data. Our robust aging-stage classifier STageR can be easily applied to other datasets, even when a small proportion of cluster cytosines is covered.

## Discussion

The widespread successful application of DNA-methylation aging clocks demonstrates an intricate link between a critical component of the epigenome and the aging process in principle. By design, however, methylation clocks are typically geared to predict a sample's chronological age based on methylation data from as few CpGs as possible. Differences between the predicted and the true age are subsequently interpreted as an acceleration or deceleration of aging. Unfortunately, little can be learned about the underlying molecular mechanisms driving the speed of aging from such a maximally reduced model. At first glance, one could imagine the aging epigenome as a weathering landscape exposed to constant erosion. Age-dependently and gradually increasing insufficiencies of the SAM cycle or decreasing methyl group availability, for example, could lead to a loss of CpG methylation at sites with a high DNA-methylation turnover. Clearly, linear models using methylation data of a comparably small set of such CpGs would be well-suited to reflect such a process. In this scenario, however, age-

related epigenomic change is a consequence of a more profound aging process that might not contribute to aging itself. Tuning a model's prediction performance while minimizing the necessary data at the same time thus yields the danger of missing essential aging processes. Consequently, investigating the methylome's link to aging calls for alternative analyses.

We have explored nonlinear aspects of age-related 5mC changes to narrow this critical gap. Analyzing the data of aging mouse colon tissues, we have identified multiple sets of CpGs exhibiting sudden methylation changes at two different time points. One group of sets undergoes a rapid methylation change during the early-to-midlife transition, while another group exhibits accelerated methylation changes during the mid-to-late-life transition. Interestingly, DNA methylation switches at similar time points were observed in rat peripheral blood DNA[55]. Notably, the division of the lifespan into three stages is already supported by the raw methylation data (Fig. 1c). Our data goes in line with a digital aging hypothesis which views aging as a process consisting of discrete steps resulting from mechanisms showing variation in rate during lifespan[56]. The existence of transitions between discrete stages reveals the more controlled, or even programmed, nature of epigenetic aging and opens questions about the

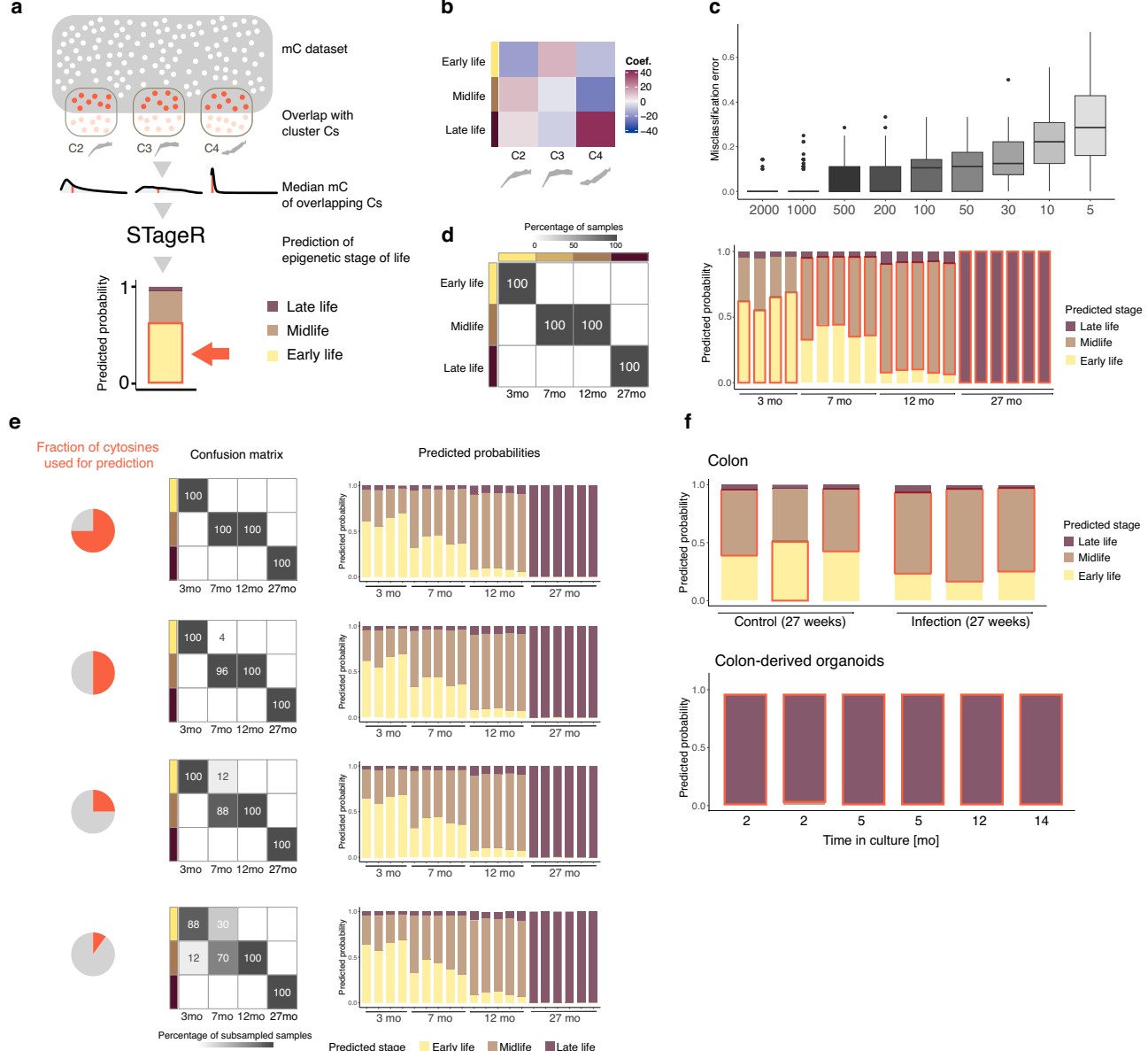

**Fig. 6 | STageR - cluster-based epigenetic stage classifier. a** Workflow of the epigenetic stage of life prediction using STageR. Firstly, the overlap between the query dataset and the nonlinear CpG clusters is identified. Median DNA methylation of overlapping CpGs is calculated, dramatically reducing the feature space's dimensionality to three. STageR performs a multinomial logistic elastic net regression and assigns probabilities to the three epigenetic stages of life. The stage with the highest probability is highlighted in red in the bar plots. **b** Mean β-coefficients in the multinomial logistic regression of STageR for clusters in the model per life stage. The bottom panel shows the associated Z-score trajectories. **c** Misclassification error in tenfold cross-validation models with a subsampled number of cytosines in each cluster (x-axis), $n = 100$ for each box plot. For the box plots, the center line shows the median, the box limits show the first and third quartiles, and the upper and lower whiskers extend from the hinge to the largest or the lowest value no further than 1.5× the interquartile range (IQR) from the hinge. **d** STageR prediction for validation dataset. Left: Confusion matrix of predicted life stages (y-axis) for validation samples from four distinct age groups (x-axis). Right: Predicted probabilities for life stages (y-axis) for all validation samples (x-axis). Red boxes are drawn around the life stage with the maximum probability. **e** STageR prediction for subsampled data. Confusion matrices and predicted probabilities when using 75, 50, 25, and 10% (from top to bottom) randomly sampled cytosines from each cluster. **f** STageR predictions for publicly available datasets[45,46]. Source data are provided as a Source Data file.

regulation and consequences of these abrupt changes. It also indicates that essential insights into the nature of aging may be missed when comparing only two ages.

The identified epigenetic transitions may have functional consequences. CpGs following the early-to-mid-life transition are strongly enriched in binding sites of Gata1, a critical transcription factor in the development of eosinophils[57]. These immune system cells were reported to play an important role in intestinal homeostasis, including the maintenance of barrier function[58]. Interestingly, the concentration of bacterial endotoxin, a marker of intestinal permeability, in mouse

plasma increases markedly between 2 and 15 mo[59]. Hence, it is likely that early-to-mid-life transition changes affect the gut barrier.

In contrast, changes observed during the mid-to-late-life transition may affect the stem cell pool. It is well known that the function of intestinal stem cells (ISCs) declines with age (reviewed in ref. 60), but the underlying molecular processes are not well understood. We identify age-dependent nonlinear DNA methylation dynamics in regions important for ISCs function, such as bivalent chromatin and binding sites of PRC2 complex components. PRC2 regulates ISCs in adult mice by the maintenance of bivalent promoters[61], and its deletion

results in the loss of ISCs[62]. Moreover, genes overlapping CpGs following mid-to-late-life transition are associated with stem cell exhaustion, an important hallmark of aging. Hence, it is tempting to speculate that the nonlinear methylation changes are linked to the decline of ISCs function in aging. Although changes in methylation levels during mid-to-late-life transition are very subtle, they may have profound consequences if the ISCs population is affected. Strikingly, gene expression of Cdx1 and Ehf, transcription factors crucial for colonic differentiation[63], follow a nonlinear trajectory with drops in expression coinciding with early-to-mid-life and mid-to-late-life transitions (Supplementary Fig. 8). Moreover, levels of apoptosis in the intestinal crypts after low-dose irradiation increase considerably in 29 mo mice compared to 18 mo and younger individuals[64], indicating that after mid-to-late-life transition, ISCs are more sensitive to damage.

Furthermore, our gene set enrichment analyses consistently show strong associations between genes affected by nonlinear 5mC changes and the nervous system. Interestingly, mid-to-late-life transition, starting after 15 months, agrees well with the timing of the loss of enteric neurons, whose number stays stable for the first 16 months of life and starts to decline afterward[2]. Thus, nonlinear DNA methylation changes may affect not only ISCs but also neural progenitor cells. One supporting evidence for this is the considerable increase in transcription and methylation of *Dmrta2* during the mid-to-late-life transition (Fig. 4c). Dmrta2 controls the cell cycle of neural progenitor cells, and its increase suppresses differentiation[35], which results in impaired neurogenesis. Furthermore, the motility of the colon, which is under the control of the enteric nervous system, undergoes changes coinciding with the timing of the mid-to-late-life transition. For instance, the total transit time is significantly prolonged in the colon from 24 mo mice compared to 18 mo and younger individuals[65], which goes in line with the age-dependent changes in 5-HT signaling, a major regulator of colonic motility[66]. The enteric nervous system (ENS) communicates extensively with the central nervous system (CNS). Thus, it may not be surprising that cognitive decline in mice also occurs from the 15th month of age onwards[67]. Interestingly, many neurodegenerative diseases are accompanied by gastrointestinal symptoms, which can even manifest earlier than dysfunctions in the CNS (reviewed in ref. 68). Moreover, the human ENS was shown to express risk genes for extraintestinal diseases[15]. Therefore, future analyses of transitions between epigenetic stages in the intestine may shed light on the aging process of other organs as well.

Based on nonlinear DNA methylation trajectories, we built an epigenomic age classifier. Our model STageR predicts the epigenetic stage of life-based on median methylation values of the nonlinear methylation clusters. The utility of our model may be counterintuitive given the existence of highly accurate epigenetic clocks predicting chronological age. However, depending on the usage context, a classifier allowing for dividing the aging process into stages may be more practical. To date, biological markers stratifying aging into clear phases are lacking. We propose that nonlinear DNA methylation trajectories may be used to stage the aging process. In contrast to current epigenetic clocks, which convey little information on underlying molecular processes, our set of cytosines is highly enriched for specific features. Moreover, STageR overcomes a major technological bottleneck of current epigenetic clocks, which rely mostly on a predefined set of cytosines and can be used predominantly with data from the array technology. The set of loci used by STageR is highly redundant, allowing for input data flexibility and usage of WGBS or RRBS datasets. STageR lays a foundation for future developments in epigenetic age prediction.

The use of the RRBS protocol, which enriches CpG-rich regions, enabled us to achieve high coverages, facilitating the cost-effective identification of modest methylation changes. In turn, our analysis is therefore biased towards CpG-rich regions. The bias likely influences which trajectories are detectable, e.g., because CpG-rich regions such

as CpG islands are frequently hypermethylated. Also, CpG-rich regions are frequently located in the vicinity of genes. Thus, trajectories specific to inter-genic regions are less likely to be captured. In the future, it will be interesting to employ whole genome bisulfite sequencing (WGBS) to profile DNA methylation across the entire genome. To conclude, we detect major nonlinearities in the progress of aging on the level of DNA methylation and propose a molecular basis as well as a tool for staging the aging process in the mouse colon. Characterizing differences between stages resulting from abrupt methylation changes is an exciting area for future investigation.

## Methods
### Animals
Male C57BL/6J/Ukj mice were bred in the Central Experimental Animal Facility (ZET) at Jena University Hospital, Jena, Germany. Mice were maintained at $22 \pm 2$ °C on a 14 h/10 h day-night cycle and at a relative humidity of $55 \pm 10\%$. Mice had unlimited access to water and food (ssniff mouse V1534-300, ssniff Spezialdiäten GmbH, Soest, Germany). Mice at ages 3 ($n = 16$), 9 ($n = 16$), 15 ($n = 16$), 24 ($n = 17$), and 28 ($n = 18$) months were used ($n = 83$ total). All studies were performed in strict compliance with the recommendations of the European Commission for the protection of animals used for scientific purposes and with the approval of the local government Thüringer Landesamt für Verbraucherschutz, Germany (license TVA 02-024/15). Experiments are in accordance with the ARRIVE guidelines. This study was performed on male mice only. The inclusion of both sexes was not possible because of limited resources.

### Animals used for validation experiment
Animal experiments were approved by the state government of Thuringia under the animal experiment license FLI-17-024 and FLI-19-012. Male C57BL6/J mice were obtained from The Jackson Laboratory. The mice were kept in individually ventilated cages (IVCs) under Specific Pathogen Free (SPF) conditions with a 12 h/12 h dark/light cycle at a temperature of 20 °C and a relative humidity of 55% according to the directives of the 2010/63/EU and GV SOLAS. Animals were sacrificed at the age of 3 ($n = 4$), 7 ($n = 5$), 12 ($n = 5$), and 27 ($n = 6$) months.

### DNA and RNA extraction
For the extraction of genomic DNA and RNA, the mice were sacrificed by cervical dislocation. Subsequently, the colon was removed, rinsed with PBS, and cut lengthwise. After that, the tissue was divided lengthwise again and cut in the middle. The tissue was snap-frozen and the proximal section of the colon was used either for DNA or RNA extraction. Genomic DNA was isolated with the DNeasy Blood & Tissue Kit (Qiagen, Cat.no: 69504). For RNA isolation, all samples were homogenized in QIAzol Lysis Reagent (Qiagen, Cat.no: 79306), and 0.2 volumes chloroform (Sigma, Cat.no: C2432) were added. Following phase separation, the aqueous phase was transferred into a fresh tube, then 0.16 volumes of NaAc (2 M, pH 4.0) (Sigma, Cat.no: S7670) and 1.1 volumes isopropanol (Sigma, Cat.no: I9516) were added. The RNA was precipitated over night at −20 °C. After centrifugation, the pellet was washed with 75% ethanol (Merck, Cat.no: 1.00983.2500). Total RNA was re-suspended in water (Invitrogen; Cat.no: 10977-035) and stored at −80 °C until use.

### RRBS library preparation and sequencing
Reduced representation bisulfite sequencing (RRBS) libraries were prepared and sequenced at the CCGA Sequencing platform (Competence Center for Genomic Analysis - Kiel, Germany), using an in-house protocol. In brief, 200 ng of DNA was fragmented via a 5 h MspI-digest (New England Biolabs, Cat.no: R0106M), followed by TruSeq adapter ligation (Illumina, Cat.no: 20020595) and subsequent bisulfite conversion following the manufacturer's protocol (EZ DNA Methylation Gold Kit, Zymo Research, Cat.no: D5005). In order to assess bisulfite

conversion efficiency, spike-in control oligos from a TrueMethyl kit (CEGX) were added prior to bisulfite conversion at a concentration of 0.4% (w/w). The bisulfite converted DNA was amplified in a 19-cycle PCR with Pfu Turbo Cx Hotstart DNA Polymerase (Agilent Technologies, Cat.no: 600410), and the resulting libraries were quality checked using 4200 TapeStation (Agilent Technologies, Cat.no: 5067-5584). Finally, the RRBS-libraries were pooled and sequenced in paired-end mode with 50 bp read lengths (S2 reagent kit, Illumina, Cat.no: 20028316) on a NovaSeq 6000 machine (Illumina).

### Validation samples

For sequencing of validation samples, Illumina's next-generation sequencing methodology[69] was used. In detail, genomic DNA was quality-checked and quantified using the 4200 TapeStation instruments in combination with the Genomic DNA ScreenTape (both Agilent Technologies). Libraries were prepared from 100 ng of input material using Ovation RRBS Methyl-Seq with TrueMethyl oxBS (Tecan, Cat.no: 0553-32). In detail, only the bisulfite part of the protocol was followed, oxidation of DNA was not done, and samples were treated as MOCK oxBS samples according to the manufacturer's instruction. Quantification and quality checks of libraries were done using the 4200 TapeStation and D5000 ScreenTapes instruments (both Agilent Technologies). Libraries were pooled and sequenced on a NovaSeq 6000 (Illumina) using S1 100-cycle reagents (Illumina, Cat.no: 20028319). The system was running in 101 cycle/single-end/standard loading workflow mode. Sequence information was converted to FASTQ format using bcl2fastq v2.20.0.422.

### RRBS preprocessing

An in-house RRBS analysis pipeline was applied to obtain significantly differentially methylated regions within the mouse genome GRCm38 retrieved along with its gene annotation from Ensembl v102. The bisulfite conversion was measured across all cytosines of spike-in control sequences and was found to have a mean efficiency of 99.7%. According to inspections made from FastQC v0.11.9 (https://www.bioinformatics.babraham.ac.uk/projects/fastqc) reports, low-quality sequences and Illumina universal adapter-, as well as poly mononucleotide content was removed from the 3' end of the reads using Cutadapt v2.10[70] (quality cutoff 20), thereby clipping end-repaired MspI cutting sites. The preprocessed data was then aligned to the reference genome using the bisulfite treatment aware mapping software segemehl v0.3.4[71,72] with an adjusted accuracy (95%). Mappings were filtered by Samtools v1.12[73] for uniqueness and properly aligned mate pairs. Overlaps with first mate sequences were trimmed off from the second mate utilizing BamUtil clipOverlap v1.0.14[74]. Afterward, methylation rates were reported for all cytosines within a CpG context that have a read coverage of at least 10. For this purpose, the data-adaptive variant caller haarz v0.3.0[75] was used. One sample was excluded from downstream analysis as it was identified as an outlier in the principal component analysis (Supplementary Fig. 1a).

### RRBS data analysis

For further analysis, we included CpGs that were covered by at least eight biological replicates in all age groups (1,535,823 Cs in total), hereinafter referred to as background cytosines.

Principal component analysis was calculated using prcomp function in the R stats package. Samples were clustered using 10,000 randomly selected background CpGs using the hierarchical complete linkage clustering method and Euclidean distance of methylation levels.

For each pairwise combination of age groups (e.g. 3 months vs. 9 months, 3 months vs. 12 months, etc.), BEDTools v2.29.2[76] was utilized to compile matrices of methylation rates. Afterward, significantly differentially methylated regions during aging (aDMRs) were called using metilene v0.2.8[77] ($q$ value ≤0.05 at a minimum of eight required

CpGs per DMR and eight data points per condition and CpG). All aDMRs were then joined, resulting in 3981 aDMRs. CpGs overlapping the aDMRs were extracted, resulting in 84,693 CpGs, hereinafter referred to as aDMRs CpGs.

### Clustering

All aDMR CpGs were clustered based on the Z-scores of the age group mean methylation rate by clust v1.12.0[17] with four iterations in total. After the first clustering round, 31% of aDMR-associated CpGs remained unclustered. Hence, three additional rounds of clustering in an iterative manner were performed resulting in 19 clusters comprising 94% of all aDMR CpGs.

### Annotations

aDMRs and aDMR CpGs were annotated to genes and genomic features using annotatr v1.20.0[78] and TxDb.Mmusculus.UCSC.mm10.knownGene v3.10.0 R packages. The promoter region was set as 1 kbp upstream from the transcription start site (TSS). Random CpGs were sampled from background CpGs.

### Enrichment analyses

Tracks with regulatory elements from ORegAnno[79] and ENCODE chromatin states from mouse intestine at P0[36] were downloaded from UCSC Table Browser on 3 June 2019 and 25 May 2021, respectively. Overlap between CpGs undergoing different trajectories in aging and genomic features were performed with GenomicRanges v1.46.1[80]. Enrichments are presented as odds ratio (OR), and the $p$ value was calculated with Fisher's exact test using R fisher.test function (two-sided) and adjusted with the Benjamini−Hochberg method. For OR calculation aDMR CpGs were compared to the mean of ten randomized datasets of the same size drawn from background CpGs. ComplexHeatmap v2.10.0 was used for hierarchical clustering of age-dependent trajectories based on the calculated OR. Enrichment in tissue-specific genes was analyzed with TissueEnrich v1.14.0[81].

Overlapping genes with aDMR CpGs and background CpGs were found using annotatr v1.20.0 when the CpGs overlapped the annotation of gene promoters, exons, introns, 5'UTRs or 3'UTRs in mm10 genome. GO enrichment analysis for molecular functions and biological processes was performed on all genes overlapping aDMR CpGs against genes overlapping background CpGs using clusterProfiler v4.2.2.

### Hallmark of aging enrichment analysis

Human genes associated with Hallmarks of Aging were obtained from[8]. Their mouse orthologs were found using Ensembl gene orthologs in biomaRt v2.50.3 and org.Mm.eg.db v3.14.0. Genes overlapping with CpGs in clusters C1−C5 were found using annotatr v1.20.0 and further filtered for genes with at least six CpGs from the corresponding cluster. Enrichments were calculated using GeneOverlap v1.30.0 for the set of each hallmark genes found also associated with the particular cluster against all genes that are associated with the particular cluster and have a human ortholog. Enrichments are presented as odds ratios (OR).

### Functional gene set enrichment analyses

Cytoscape v3.8.2 with a plug-in ClueGO v2.5.8[82] was used for the gene ontology analysis and its visualization with the following settings: GO biological process, custom reference set (genes overlapping with background CpGs), right-sided hypergeometric test with Bonferroni step down correction, $p$ value ≤0.05.

Ingenuity Pathway Analysis v70750971 (IPA, QIAGEN)[83] was used for canonical pathway and potential upstream regulator analysis. Genes overlapping with background CpGs were used as a custom reference set, and $p$ values were adjusted with the Benjamini−Hochberg method.

## RNA-Seq library preparation, sequencing, and data processing

RNA-sequencing libraries were prepared with the Illumina TruSeq Stranded mRNA Library Preparation kit according to the vendor's protocol. The resulting libraries were sequenced on a NovaSeq 6000 machine in paired-end mode with read lengths of 100 bp. We employed cutadapt (v2.8) to remove adapter sequences from the raw reads with a minimum overlap of 3 bp and a maximum error tolerance for 10% mismatches (TrueSeq forward adapter = GATCGGAAGAGCACAC; TruSeq reverse complement universal adapter = GATCGGAAGAGCGTCGTG TAGGGAAAGAGTGTAGATCTCGGTGGTCGCCGTATCATT). We did additional 3′-end quality trimming for a minimum Phred-score of 25, while also trimming poly-G ends due to Illuminas two-color chemistry dark-cycle issues, with the cutadapt option --nextseq-trim=25. An additional quality filtering step was done with PrinSeq Lite (v0.20.4) with a mean read quality over all bases of at least Phred-score 15, a maximum of 8 unknown basecalls, and minimum read lengths of 20 bp. Post-QC read qualities were visually checked using FastQC (v0.11.9). Using Hisat2 (v2.1.0) software, the filtered reads were mapped against the *Mus musculus* reference genome GRCm38 released by the European Bioinformatics Institute in version 99 (February 2020). Only uniquely mapped reads were kept, employing SamTools (v1.9) -F 256 flag. Gene abundances were counted for properly mapped read pairs, taking into account strandedness information (-s 2), with the tool "featureCounts" from the subread software (v2.0.1).

## RNA-seq data analysis

TPMs for all genes were calculated as raw read counts divided by the merged gene length in kilobases (RPKs) and normalized then by the scaling factor defined as the sum of all RPK values divided by $10^6$. For each age group, genes were categorized as active (mean TPM >0) or inactive (mean TPM = 0). Active genes were further divided into expression deciles, with the lowest expressed genes in the first decile and the highest expressed genes in the tenth. Differentially expressed genes (DEGs) were assessed with the R-package DESeq2 v1.40.2[84]. All samples were loaded and processed together with a design formula accounting for the library preparation batch and the age group as a factor. Pairwise comparisons were extracted from the DESeq results object for each one of the ten possible age group combinations. Genes with *p*adj <0.1 were considered significant. DEGs were clustered based on the Z-scores of the age group mean TPM by clust v1.12.0[17] with five iterations in total. A protein–protein interaction network was built from genes undergoing the mid-to-late-life transition on both DNA methylation and gene expression levels with the STRINGApp v2.0.1 of Cytoscape[85] with a confidence score = 0.4. To test for the enrichment of genes associated with a particular methylation cluster and differentially expressed genes undergoing transition at the same time, fisher.test function in R version 4.2.2 stats package was used with all genes covered in the RRBS experiment as the joint set. To test for the enrichment of genes associated with C4 in differentially expressed genes between the same time points, all differentially expressed aging genes covered in the RRBS experiment were taken as the joint set.

## Cell type deconvolution from bulk RNA-seq data

Cell type compositions of our bulk sequencing data were estimated from RNA-Sequencing profiles via the R-package MuSiC (v1.0.0). MuSiC employs single-cell gene expression data to infer cell type compositions from bulk RNA-seq data[86]. We have obtained large intestine single-cell expression data from Tabula Muris[87] as a reference dataset for MuSiC. For cell type annotation, we chose the cell ontology classification, which was present in the Seurat-objects metadata information. The cell ontology classification offered five different cell types: 2019 epithelial cells, 964 enterocytes, 833 goblet cells, 63 brush cells, and 59 enteroendocrine cells. In this classification, epithelial cells were all undifferentiated cells either Lgr5+ (stem cell marker) or Lgr5−, while enterocytes were considered properly differentiated cells. We used the raw counts from our bulk RNA-Seq data and annotated the ENSEMBL gene IDs with gene symbols. Matching the bulk gene symbols to the single-cell gene symbols gave us an overlap of 21,003 genes. Deconvolution proportions were calculated with cell ontology class as "clusters" and single-cell data mouse id as "samples" variables. The deconvolution scores good explained variance ($R^2$-values) in the range of 0.26−0.61. Differences in the cell type compositions between the age groups were tested with Kruskal Wallis and Dunn's post hoc test, corrected for multiple testing via Benjamini−Hochberg FDR.

## Validation dataset preprocessing and analysis

The validation RRBS data preprocessing was slightly adapted to match the different library preparation protocol and single-end sequencing technique. Reads that contain a diversity adapter of length zero to three, following an MspI cutting site, were selected prior to adapter clipping, sequence quality, and content assessment. Further, aligned data was deduplicated for over-amplified PCR fragments based on unique molecular identifiers utilizing UMI-tools v1.1.1[88]. The downstream processing of the dataset was performed analogously to the initial dataset. Here, we included CpGs that were covered by at least four or 80% of biological replicates in the age group. aDMRs were again called between all combinations of age groups. The underlying CpGs were then compared with the CpGs in the aDMRs of the initial dataset using the GenomicRanges R package. To assess the similarity of the cluster trajectories, the overlapping cytosines were assigned to the corresponding cluster. Then the Euclidean distance to the cluster centroids per age category were calculated for all these cytosines. 70% of the closest CpGs with minimal distance to the cluster centroid were selected for the visualization in Fig. 5. The cubic smooth spline fit was calculated based on the Z-scores in the original dataset using smooth.spline with 5 degrees of freedom. The same spline fit was projected on the validation data vertically moved by the difference of the median methylation levels at the initial state (3 mo). The distribution of the methylation values was shown using box plots of mean values over all samples in the corresponding age group.

## Publicly available dataset processing and analysis

Publicly available RRBS DNA methylome experiments from colon samples upon induced intestinal inflammation (GSE163037) and colon organoids (GSE114801) were processed according to their experimental designs and library preparation methods. Datasets of the first project, derived from a paired-end library with diversity adapters, were filtered for read pairs which contain MspI cutting sites following a diversity adapter, which was subsequently clipped prior to the application of our analysis pipeline as described above. For the second project, our pipeline was adjusted to match the requirements to process reads from single-end sequenced samples, filtered for MspI cutting site sequences at their 5′ end.

## Cluster-based epigenetic clock

To build STageR (STage of AGing Estimator)−an epigenetic clock based on the methylation information in clusters, we first calculated the median methylation value in each of the nonlinear clusters C2, C3, and C4 for each sample. This data matrix consisting of three cluster values for 82 samples was used as the feature matrix in the elastic net regression. The corresponding life stage (early life = 3 mo, midlife = 9 mo and 15 mo, late-life = 24 mo and 28 mo) was used as the response variable. Multinomial logistic regression with elastic net regularization and alpha = 0.5 was used to fit the epigenetic clock. The performance of the original dataset was assessed by tenfold cross-validation with folds equally distributed over the life stages. The optimal parameter lambda was assessed through the nested tenfold cross-validation procedure. Reported coefficients for all clusters are the beta mean values from the outer tenfold cross-validation procedure which was repeated ten times to assess for possible sample biases in the folds.

The full model used for later predictions was fit using all 82 samples in the original dataset. The elastic net regression was trained and tested using glmnet v4.1.6 package, and the random folds were created using the caret v6.0.93 package.

## Subsampling from clusters

To evaluate how much information per cluster is important to successfully predict the epigenetic life stage using STageR we conducted the following subsampling strategy embedded into the tenfold cross-validation procedure. In each fold, the particular number (e.g., 500, 200, 100, 50, 20, 10, and 5) of cytosines was randomly selected in each cluster to calculate the median methylation values per cluster in the test set. The median calculation in the training set remained unchanged using all cytosines. Then, the clock was built on the training set and used for the prediction of the test set based on subsampled cytosines. The misclassification error of each prediction was assessed. The subsampling procedure was repeated ten times for each number of cytosines to obtain different cytosines per cluster.

## Clock-based prediction for the validation dataset

Medians per cluster were calculated using all cytosines overlapping one of the three clusters (C2, C3, and C4) and used for the prediction of the life stage using the full STageR model. In the subsampling procedure, only a proportion of the cytosines present in both the validation and the original dataset per cluster (e.g., 75, 50, 25, and 10%, respectively) was selected randomly. This procedure was repeated ten times to obtain different sets of cytosines. The predicted probabilities for each life stage (e.g., response) were assessed for each sample.

## Clock-based prediction for the publicly available dataset

First, cytosines overlapping the selected clusters (C2, C3, and C4) were selected using GenomicRanges v1.46.1[80] for colon samples upon induced intestinal inflammation (GSE163037) and colon organoids (GSE114801). Then, medians per cluster were calculated and used for the prediction of the life stage using the full STageR model. The predicted probabilities for each life stage (e.g., response) were assessed for each sample.

## Visualization

Plots were visualized using any of the following R packages: ggplot2 v3.4.0, ggridges v0.5.4, Gviz v1.38.4, eulerr v6.1.1., and ComplexHeatmap v2.10.0.

## Reporting summary

Further information on research design is available in the Nature Portfolio Reporting Summary linked to this article.

## Data availability

The RRBS data generated in this study have been deposited in the Gene Expression Omnibus (GEO) database under accession code GSE233734. The RNA-Seq data generated in this study have been deposited in the GEO database under accession code GSE248002. The RRBS data from colon samples upon induced intestinal inflammation used in this study are available in the GEO database under accession code GSE163037. The RRBS data from colon organoids used in this study are available in the GEO database under accession code GSE114801. Mouse reference genome GRCm38 used in this study is available at [https://ftp.ensembl.org/pub/release%2D102/fasta/mus_musculus/dna/] and [https://ftp.ensembl.org/pub/release%2D102/gtf/mus_musculus/]. Source data are provided with this paper.

## Code availability

Code for STageR is available at [https://github.com/Hoffmann-Lab/STageR][89].

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

## Acknowledgements

The project has been funded by the German Research Foundation (DFG) with the project number 418087534 to S.H., C.F., and C.K. The Carl Zeiss Foundation supported the project in the context of the IMPULS con-sortium with the project number P2019-01-006 to S.H. In addition, the work was supported by the Leibniz Research Alliance "Resilient Ageing" with the project number: LFV-2021-2-LIR to S.H. and by the European Union's Horizon 2020 research and innovation program under the Marie Sklodowska-Curie grant agreement no. 859890 (SmartAge) to O.W.W., C.F., and C.K. The Core Facility Next-Generation Sequencing and Mouse Facility of the FLI, Jena, Germany, and Competence Center for Genomic Analysis, Kiel, Germany, are gratefully acknowledged for their techno-logical support. The FLI is a member of the Leibniz Association and is financially supported by the Federal Government of Germany and the State of Thuringia. The publication of this article was funded by the Open Access Fund of the Leibniz Association and the Leibniz Institute on Aging – Fritz Lipmann Institute (FLI), Jena, Germany.

## Author contributions

M.O. and S.H. designed the study. M.O., A.v.B., S.H., L.B., T.D., and S. Flor analyzed the data and developed statistical models. L.B. and K.R. pro-cessed the raw data. M.H., C.F., and B.v.E. provided material. S. Foerste and M.H. isolated nucleic acids. S. Franzenburg performed RRBS and RNA-Seq. M.G. performed RRBS of the validation samples. The manu-script was written by M.O., A.v.B., and S.H. with critical feedback from all other authors. The study was supervised by S.H., C.F., O.W., and C.K.

## Funding

## Competing interests

The authors declare no competing interests.
