## [Peer Review File · Nature Communications]

Nonlinear DNA methylation trajectories in aging male miceREVIEWER COMMENTS

Reviewer #1 (Remarks to the Author):

Olecka and colleagues review literature to suggest that molecular aging is non-linear and they argue that most studies of DNA methylation during aging, including so-called DNA methylation clocks, employ linear models to estimate age. They use a number of innovative and creative approaches to analyze cytosine DNA methylation in different ways and report that mice show three broadly distinct phases of aging in the colon based on their modeling. They replicate these findings in independent cohorts including a public one generated by others and they create a new aging clock named STageR which can predict the 3 stages of life in mice even with minimal input of methylation sites. They find that differentially methylated regions with age are enriched in genes with roles in the cortex and neuronal signaling.

This is a provocative and creative manuscript which expands the use of DNA methylation data to extract potentially new findings relevant to organismal and certainly to colon aging. It is nicely written and the figures and legends are clear. The authors provide numerous interesting interpretation of what genes they see changing during different stages of aging and it would make the paper stronger if they would validate at least some of these with orthogonal methods as suggested below.

1. to add more functional relevance to the observed findings and to ascertain that the aDMRs are indeed functional the neuronal and glial genes mentioned in line 109-112 could be measured by single cell or single nuc sequencing or localized in gut tissue sections using in situ gene or protein expression. This would help demonstrate that the neural genes are indeed functionally expressed and change with age in the colon. It seems the authors have already generated such data (line 207)?
2. Similarly, the gene regions linked by DNA methylation to stages or transition between stages could be validated using RNA or protein measurement. Here and for point 1 above, gene expression datasets may exist in the literature that recorded expression of these genes with age.
3. what is the interpretation of the statement "Strikingly, colon organoids derived from 40 days old mice show an extremely aged methylation profile with almost full hypermethylation even after only two months of culture ". Could the authors elaborate. Without functional or at least gene or protein expression changes it seem difficult to know what the DNA methylation state means in this context.
4. What do the authors mean by "To examine the association between DNA methylation and gene expression, we performed RNA-Seq on all the samples. We found that most genes overlapping with aDMRs show weak or no RNA-Seq signals (TPM < 1)." This seems confusing as it suggests these genes are never expressed and that methylation has nothing to do with gene activity? One would expect that the genes are expressed at least in young colon and then maybe silenced with aging. In contrast, if the gene is already silenced at the youngest age analyzed and not expressed then what would be the functional biological effect relevant to aging of further methylation? Could the author provide and interpretation here?

Reviewer #2 (Remarks to the Author):

Olecka et al evaluate the DNA methylation landscape and transcriptional profiles of colon cells isolated from male C57Bl/6J mice of ages 3,9,15, 24, and 28 months, importantly not just from young and aged mice based on the assumption of linear methylation changes. They identify clustering of global DNA methylation that is indicative of non-linear aging patterns and delve more into finding CpG methylation profiles that can identify age-stages by the levels of DNA methylation at specific loci. While consistent with many other studies suggesting that there are not many DMR loci with transcriptional changes, the authors do examine a few key genes / pathways that appear relevant for

neuronal pathways as well as possible ISC regulatory genes. It is appreciated the expense and animals required limited the age groups that could be addressed in this study, but the kinetics of the age "stages" would be strengthened with additional time points – that would either support the 3 stages or suggest additional stages. This seems critical as the authors develop the epigenetic clock to define the age stage using clusters of DNA methylation with patterns that support a shift from young to middle and middle to aged profile, though other patterns of DNA hypermethylation are present in almost equal numbers- (where the shift would be from 9-15 rather than 3-9). Further it is unclear to this reviewer how the use of methylation marks with non-linear trajectories are more robust to predict functional changes over the linear groups.

Major Points:

A key point of this manuscript is that non-linear DNA methylation changes may be useful to better depict the life-stage of the colon cells- with the major transitions defined as young to mid (3-9 months), and then mid to old (15-24:28)

- 1) Are there specific functional changes in the colon that these age ranges associate with?
- 2) Using the transcriptional profiles generated from the colon cells at each of these time points, is there a similar break in the trajectories?

As the authors note, several manuscripts have recently highlighted the idea of non-linear aging, with timepoints in life with significant trajectory shifts that appear to be non-random. Indeed a recent manuscript looking at rat blood aging (doi.org/10.7554/eLife.76808) similarly defined non-linear methylation changes in blood that correlated with changes in the composition of the cell types associated with aging. Are there significant alterations in the composition of the colon cells that correlate with the age-clusters suggested by the authors? Significant functional differences?

Profiles from groups C2, C3, and C4 (total of 12,725 DMR) were used to model the chronological clock in STageR. However, it appears that clusters C7, C8 and C9 (9,624 DMR) could also have been used to identify "age-stages" with the potential inference the stage transition was between 9-15 months rather than the 3-9. If the C7-C9 clock was used, would the 7 month old validation mice then be identified as "early life" stage? Could the authors be more explicit as to why where C2-C4 chosen to reflect the stages, and why biologically this is "correct" groups for modelling aging? If more ages were added, would there perhaps be additional transitions / stages that would need to be addressed?

It would be helpful if the authors could elucidate more the utility of this methylation clock fitting colon cells into age-stage, rather than a predicted age. If it is being used to identify age-stages, with potentially different functional potential, couldn't the stage of the cells also be identified using a subset of DMRs with linear trajectories? Does this epigenetic clock have utility in other systems, or is it a colon specific clock? Could it have utility beyond showing not all age methylation changes are linear?

If age-stages are critical and methylation helps dictate these stages, could they authors speculate why loss of DNA methylation is only linear? Also, could the authors also address their thoughts on how so few of the DNA methylation changes are in intergenic regions (lines 97-99) and why there is a depletion of interCGI DMRs compared to random?

The authors highlight that many of the mid-to-late life DNA methylation transitions are at bivalent promoters- it would be helpful for perhaps a supplemental figure to be included here and could also be used to show if this enrichment was specific to methylation changes at this stage transition or if these bivalent domains were also enriched in other patterns (especially the linear gains). Also the authors should show the OD for SET domain-containing methyltransferase for information about the H3K4me.

Minor Concerns:

Figures are difficult to read due to small text- and figure labels in 2B are unclear.

REVIEWER COMMENTS

Reviewer #1 (Remarks to the Author):

Olecka and colleagues review literature to suggest that molecular aging is non-linear and they argue that most studies of DNA methylation during aging, including so-called DNA methylation clocks, employ linear models to estimate age. They use a number of innovative and creative approaches to analyze cytosine DNA methylation in different ways and report that mice show three broadly distinct phases of aging in the colon based on their modeling. They replicate these findings in independent cohorts including a public one generated by others and they create a new aging clock named STageR which can predict the 3 stages of life in mice even with minimal input of methylation sites. They find that differentially methylated regions with age are enriched in genes with roles in the cortex and neuronal signaling.

This is a provocative and creative manuscript which expands the use of DNA methylation data to extract potentially new findings relevant to organismal and certainly to colon aging. It is nicely written and the figures and legends are clear. The authors provide numerous interesting interpretation of what genes they see changing during different stages of aging and it would make the paper stronger if they would validate at least some of these with orthogonal methods as suggested below.

1. to add more functional relevance to the observed findings and to ascertain that the aDMRs are indeed functional the neuronal and glial genes mentioned in line 109-112 could be measured by single cell or single nuc sequencing or localized in gut tissue sections using in situ gene or protein expression. This would help demonstrate that the neural genes are indeed functionally expressed and change with age in the colon. It seems the authors have already generated such data (line 207)?

RESPONSE: *We thank the reviewer for this valuable suggestion. Following the reviewer's suggestion, we have evaluated the expression of neural and glial cell genes in the bulk RNA-Seq data, which we added to Supplementary Figure 1cd. There is a trend toward downregulating neural genes and upregulating glial genes during aging. Unfortunately, single-cell data has not been generated in the context of this study. We have revised the sentence in line 207 to avoid any confusion.*

2. Similarly, the gene regions linked by DNA methylation to stages or transition between stages could be validated using RNA or protein measurement. Here and for point 1 above, gene expression datasets may exist in the literature that recorded expression of these genes with age.

RESPONSE: *We certainly agree with the reviewer that gene methylation changes associated with a response on RNA or protein level yield much more substantial evidence for the regulatory impact of the observed epigenomic changes. We now integrate our methylation data with expression data in response to your and the other reviewer's comments. We can now show that genes associated with CpGs undergoing a nonlinear mid-to-late-life transition (C4) are enriched in differentially expressed genes between 15 and 24 months (pvalue = 0.007, Fisher's exact test). Specifically, 13% of C4 genes are also differentially expressed between 15 and 24 months. On the other hand, 93 (5%) of the genes associated with nonlinearly changing CpGs during the early-to-mid-life transition are differentially expressed between 3 and 9 months of age. Moreover, as detailed in the response to point 4, we identified genes following the same trajectories on methylation and gene expression levels.*

While we are surprised by the strong enrichment of C4 genes in differentially expressed genes, we would like to stress that changes still measurable in the methylome may only have a transient effect on RNA expression, not visible at the sampling time.

In this respect, we would like to note that 5mC may exert different effects on transcription depending on sequence context, e.g., DNA methylation of the gene body is associated with increased gene activity¹. Interestingly, even within the promoter region, 5mC may modulate transcriptional control differently. Specifically, it was shown that promoter methylation may not affect transcription at all or result in increased

gene activity, possibly by eviction of methyl-sensitive transcriptional repressors^{2,3}. Moreover, DNA methylation change may lead only to a transient response on the level of RNA and protein, which may not align with the timing of material collection.

3. what is the interpretation of the statement “Strikingly, colon organoids derived from 40 days old mice show an extremely aged methylation profile with almost full hypermethylation even after only two months of culture “. Could the authors elaborate. Without functional or at least gene or protein expression changes it seem difficult to know what the DNA methylation state means in this context.

RESPONSE: The cited phrase specifically refers to the DNA methylation status at CpGs following nonlinear methylation trajectories in aging. Figure 3 shows that CpGs from nonlinear clusters C2, C3, and C4 exhibit hypermethylation with age. While median methylation in C2 and C3 in very old mice reaches 50%, organoids gain almost complete methylation at these loci. Similarly, in the case of cluster C4, organoids gain considerably more methylation compared to normal aging. The authors of the study in which the organoid data was generated found that this aging-like hypermethylation signature of the organoids predisposes them to malignant transformation⁴. Moreover, most promoter hypermethylation events in colorectal cancer occur at bivalent promoters⁵. As CpGs undergoing mid-to-late-life transition (C4) are highly enriched in bivalent promoters, we speculate that the observed hypermethylation may play a role in the development of colorectal cancer. Interestingly, 92 C4 genes are upregulated, and 27 downregulated when comparing gene expression between 15 and 24 mo (see Figure below). This indicates that some non-conventional forms of methylation-mediated promoter control (see response to point 2) or decoupling of methylation and transcription occurred at these sites.

4. What do the authors mean by “To examine the association between DNA methylation and gene expression, we performed RNA-Seq on all the samples. We found that most genes overlapping with aDMRs show weak or no RNA-Seq signals (TPM < 1).” This seems confusing as it suggests these genes are never expressed and that methylation has nothing to do with gene activity? One would expect that the genes are expressed at least in young colon and then maybe silenced with aging. In contrast, if the gene is already silenced at the youngest age analyzed and not expressed then what would be the functional biological effect relevant to aging of further methylation? Could the author provide an interpretation here?

RESPONSE: We agree with the reviewer that our summary on the relation between methylation and expression may have caused some confusion. In response to this comment and the questions of Reviewer 2, we reanalyzed the gene expression dataset and now provide a new section on the integration of methylation and gene expression data (see “Nonlinear gene expression trajectories in aging,” Figure 4, Supplementary Fig. 3).

First, we found that most genes showing a change in DNA methylation with age are active (see Methods). Secondly, genes exhibiting early-to-mid-life changes on the level of DNA methylation show a higher transcriptional activity compared to those showing a mid-to-late life transition (Supplementary Fig. 3b). Finally,

genes overlapping with mid-to-late-life transition CpGs (C4) are expressed at a lower level but are still active (Supplementary Fig. 3b). The latter is particularly exciting, as C4 genes enrich bivalent promoter marks and are thought to be silent in differentiated cells.

Furthermore, we identified nonlinear gene expression trajectories and found that the most prominent cluster represents genes undergoing mid-to-late-life transition. Interestingly, genes following the same transition at the level of DNA methylation and transcription are highly relevant for colon biology and are known regulators of the enteric nervous system, intestinal barrier, and colorectal cancer (Fig. 4d). Therefore, this finding further strengthens our initial observation that nervous processes are particularly affected by nonlinear transitions.

To address the question regarding the function of methylation of already silent genes, we would like to use the example of bivalent chromatin regions. Bivalent promoters are characterized by the presence of both activating (H3K4me3) and repressing (H3K27me3) histone marks and lack of DNA methylation. This state is associated with gene repression, but the absence of DNA methylation keeps the promoters accessible. Bivalent promoters are often associated with critical developmental genes that become activated during differentiation, guiding the fate of cells. Some bivalent promoters maintain this state in differentiated cells, which is thought to have a function for in-vivo de-differentiation and regeneration processes.

The acquisition of DNA methylation shifts the bivalent genes from reversible to irreversible repression. This phenomenon is frequently observed in cancer and is thought to impair the differentiation potential of cells, conferring malignant properties. To sum up, hypermethylation of an already silent gene may hinder its responsiveness to transient biological cues such as differentiation.

Reviewer #2 (Remarks to the Author):

Olecka et al evaluate the DNA methylation landscape and transcriptional profiles of colon cells isolated from male C57Bl/6J mice of ages 3,9,15, 24, and 28 months, importantly not just from young and aged mice based on the assumption of linear methylation changes. They identify clustering of global DNA methylation that is indicative of non-linear aging patterns and delve more into finding CpG methylation profiles that can identify age-stages by the levels of DNA methylation at specific loci. While consistent with many other studies suggesting that there are not many DMR loci with transcriptional changes, the authors do examine a few key genes / pathways that appear relevant for neuronal pathways as well as possible ISC regulatory genes. It is appreciated the expense and animals required limited the age groups that could be addressed in this study, but the kinetics of the age “stages” would be strengthened with additional time points – that would either support the 3 stages or suggest additional stages. This seems critical as the authors develop the epigenetic clock to define the age stage using clusters of DNA methylation with patterns that support a shift from young to middle and middle to aged profile, though other patterns of DNA hypermethylation are present in almost equal numbers- (where the shift would be from 9-15 rather than 3-9). Further it is unclear to this reviewer how the use of methylation marks with non-linear trajectories are more robust to predict functional changes over the linear groups.

Major Points:

A key point of this manuscript is that non-linear DNA methylation changes may be useful to better depict the life-stage of the colon cells- with the major transitions defined as young to mid (3-9 months), and then mid to old (15-24:28)

1) Are there specific functional changes in the colon that these age ranges associate with?

RESPONSE: We agree with the reviewer that the question on potential physiological impacts is very exciting. Unfortunately, there are not many publications that we could use to address this question in detail since the vast majority of studies compare only two time points: young and old. As mentioned in the manuscript, a sudden loss of neurons in the mouse colon between 16 and 20 was observed ⁶. This may affect colon motility. Interestingly, it was shown that total colonic transit time ⁷ and 5-HT signaling ⁸, a major regulator of colonic motility, undergo changes coinciding with the mid-to-late-life transition. Moreover, levels of apoptosis in the intestinal crypts after low-dose irradiation increase considerably in 29-month-old mice compared to 18-month-old mice and younger individuals ⁹, indicating that intestinal stem cells are more sensitive to damage after mid-to-late-life transition. In addition, a substantial decrease in male mouse fertility was observed during the mid-to-late-life transition (18 - 24 mo) ¹⁰.

CpGs with a methylation that follows an early-to-mid-life transition are strongly enriched in binding sites of *Gata1*, a critical transcription factor in the development of eosinophils ¹¹. These immune system cells were reported to play an essential role in intestinal homeostasis, including maintaining the barrier function ¹². Interestingly, the concentration of bacterial endotoxin, a marker of intestinal permeability, increases markedly between 2 and 15 months in mouse plasma ¹³. Hence, it is likely that early-to-mid-life transition changes affect the gut barrier.

We updated our discussion accordingly and better highlighted these exciting findings.

2) Using the transcriptional profiles generated from the colon cells at each of these time points, is there a similar break in the trajectories?

In response to this comment and the comment of Reviewer 1 (point 4), we reanalyzed gene expression dataset and performed clustering of differentially expressed genes, using an analogous strategy as in the case of DNA methylation analysis. The biggest cluster in gene expression analysis (CE1) contains genes showing a non-linear increase in transcriptional activity between 15 and 24 mo, i.e., during mid-to-late-life transition. There are two other clusters corresponding to transitions defined based on CpG methylation, i.e. genes showing downregulation during mid-to-late-life transition (cluster CE6) and as well as genes downregulated during early-to-mid-life transition (CE5).

As the authors note, several manuscripts have recently highlighted the idea of non-linear aging, with timepoints in life with significant trajectory shifts that appear to be non-random. Indeed a recent manuscript looking at rat blood aging (doi.org/10.7554/eLife.76808) similarly defined non-linear methylation changes in blood that correlated with changes in the composition of the cell types associated with aging. Are there significant alterations in the composition of the colon cells that correlate with the age-clusters suggested by the authors? Significant functional differences?

We thank the Reviewer for bringing the manuscript to our attention. It indeed supports timing of methylation transitions observed in our study. We added following sentence to discussion:

“Interestingly, DNA methylation switches at similar time points were observed in rat peripheral blood DNA”.

Unfortunately, single-cell data has not been generated in the context of this study. Thus, we performed cell type deconvolution analysis on the bulk RNA-Seq data and did not detect major shifts in cell type composition. We added following sentence to Results, Nonlinear DNA methylation trajectories in aging:

“Importantly, DNA methylation transitions are not accompanied by considerable shifts in cell type composition (Supplementary Fig. 4).”.

Functional changes in the colon are listed in response to point 1.

Profiles from groups C2, C3, and C4 (total of 12,725 DMR) were used to model the chronological clock in STager. However, it appears that clusters C7, C8 and C9 (9,624 DMR) could also have been used to identify “age-stages” with the potential inference the stage transition was between 9-15 months rather than the 3-9. If the C7-C9 clock was used, would the 7 month old validation mice then be identified as “early life” stage? Could the authors be more explicit as to why where C2-C4 chosen to reflect the stages, and why biologically this is “correct” groups for modelling aging? If more ages were added, would there perhaps be additional transitions / stages that would need to be addressed?

RESPONSE: We thank the reviewer for their critical but helpful evaluation of our results. While we are understandably hesitant to claim “the biologically correct selection,” our choice of early-to-mid-life (3-9 mo) and mid-to-late-life (15-24 mo) clusters is underpinned by several important observations:

i) Hierarchical clustering of raw methylation data (Fig. 1c) already results in the division of samples into three main groups: 1) 3 mo, 2) 9 and 15 mo, and 3) 24 and 28 mo. This very partition coincides with the separation induced by non-linear methylation trajectories, indicating critical early-to-mid-life and mid-to-late-life changes.

ii) Early-to-mid-life (C2 and C3) as well as mid-to-late-life (C4) methylation clusters, together with C1 and C6 were detected during the first clustering round with *clust*¹⁴. The timely detection indicates that the CpGs in these clusters exhibit a comparably strong signal.

iii) Gene expression trajectories (Fig. 4a) support the timing of transitions identified on the level of DNA methylation. Specifically, cluster CE1 (the largest cluster) and CE6 support mid-to-late-life transition and CE5 supports early-to-mid-life transition.

Thanks to the intriguing suggestion of the reviewer, we were curious how such a clock trained using the clusters C7-C9 would perform. We have trained the STager similarly but using the clusters C7, C8, and C9 instead of C2-C4. However, this clock performed considerably worse on the training and the validation data set. The main reason for the diminished performance was the difficulty in adequately learning the early-age stage signature (see confusion matrix calculated based on cross-validation of the training set). Further, all 3 months-old samples from the validation data were wrongly classified into the late stage (see the barplot with predicted probabilities). This analysis further supports our choice of early-to-mid-life (3-9 mo) and mid-to-late-life (15-24 mo) clusters.

In any case, we would like to reiterate that we do not claim that the early-to-mid-life and mid-to-late-life transitions are the only methylation transitions during aging. To avoid any such misunderstanding, we have updated the manuscript accordingly.

A detailed analysis of the minor clusters may provide interesting insights. However, due to the relatively low number of CpGs in alternative clusters, it would be necessary to generate a dataset covering the whole genome, e.g., via WGBS, to allow a meaningful characterization. Generating such a data set with more time points will undoubtedly be the focus of our future work.

It would be helpful if the authors could elucidate more the utility of this methylation clock fitting colon cells into age-stage, rather than a predicted age. If it is being used to identify age-stages, with potentially different functional potential, couldn't the stage of the cells also be identified using a subset of DMRs with linear trajectories? Does this epigenetic clock have utility in other systems, or is it a colon specific clock? Could it have utility beyond showing not all age methylation changes are linear?

RESPONSE: We thank the Reviewer for this thought-provoking comment. Nonlinear methylation changes are very intriguing and raise many exciting questions on the regulation of the transitions and their consequences. It is unclear whether there is any form of regulatory relationship between CpGs of genes affected by linear and nonlinear trajectories. To address this question, we included the linear cluster C1 into STageR to see whether it is selected as an essential feature for predicting the age stages (see figure below). The coefficients of C1 in the multinomial regression were estimated to be zero for the early and midlife stages and only slightly positive (beta = 3.4) for the late life stage. However, the estimated coefficient of C4 for late-life stage prediction was substantially higher (beta = 8.2), indicating a stronger association of the age stages with the nonlinear clusters.

In terms of tissue specificity, at this point, we can only demonstrate the utility in the colon confidently. Unfortunately, the number of suitable, publicly available RRBS methylation data sets is still limited. However, we were able to obtain and analyze RRBS data from the liver and lung by Meer et al.¹⁵. After removing outliers, we analyzed 18 liver samples and 17 lung samples of ages 6, 12, 20, and 30 months. The overlap of aDMR-CpGs in our colon data and the data from Meer et al. was limited (766 CpGs in lung, 5875 in liver). We applied STageR for stage prediction using all overlapping CpGs in clusters C2, C3, and C4. The predictions for lungs of ages 6, 12, and 20 months were accurate. All 6-month-old mice were classified into the early life stage; half of the 12-month-old were also classified into the early life stage and the other half into the midlife stage, confirming that a 12-month-old mouse is on the border between these two stages. All 20-month-old mice were correctly classified in the midlife stage. However, all old mice of age 30 were wrongly classified into the midlife stage. This misclassification might be caused by the slight overlap of sequenced CpGs in cluster C4, essential for predicting the late life stage. Similarly, most liver samples were correctly classified into the life stages. However, half of the old mice aged 30 months were wrongly classified as the midlife stage.

Confusion matrix for lung and liver samples from Meer et al. ¹⁵. The numbers show the percentage of samples classified into the corresponding life stage (in rows).

Therefore, we have limited evidence that some nonlinearly methylated CpGs reflect global and tissue-specific mechanisms at the same time. This finding highlights the necessity to analyze more tissues for non-linear transitions and to distinguish between tissue-specific and tissue-unspecific changes.

Our manuscript demonstrates that several critical pathways are affected by non-linear aging. Thus, at this point, the clock provides a simple way to classify the aging status of these biologically relevant molecular mechanisms. For instance, the prediction of the mid-to-late-life transition critically depends on the methylation of cluster C4, a cluster strongly enriching bivalent promoters. Notably, hypermethylation of bivalent promoters has been proposed as a potential marker of carcinogenesis in aging ⁵. Thus, one might speculate that a C4-dependent “misclassification” could have some prognostic value. Of course, more data for more time points is needed to develop the clocks in this direction.

If age-stages are critical and methylation helps dictate these stages, could they authors speculate why loss of DNA methylation is only linear?

RESPONSE: In aging, global DNA hypomethylation and hypermethylation of CpG islands were reported ¹⁶. Our study used reduced representation bisulfite sequencing (RRBS), specifically designed to capture CpG-rich regions ¹⁷. Since CpG-rich regions are known to be frequently hypermethylated, we identified only a low number of hypomethylated CpGs.

We added the following paragraph in the discussion to highlight the advantages and limitations of RRBS:

“The use of the RRBS protocol, which enriches CpG-rich regions, enabled us to achieve high coverages, facilitating the cost-effective identification of modest methylation changes. In turn, our analysis is therefore biased towards CpG-rich regions. The bias likely influences which trajectories are detectable, e.g., because CpG-rich regions such as CpG islands are frequently hypermethylated. Also, CpG-rich regions are frequently located in the vicinity of genes. Thus, trajectories specific to inter-genic regions are less likely to be captured. In the future, it will be interesting to employ whole genome bisulfite sequencing (WGBS) to profile DNA methylation across the entire genome.”

The mechanisms governing age-related DNA methylation changes are still not well understood as different mechanisms can cause the loss and gain of DNA methylation. The linear loss of methylation in aging may primarily be caused by passive processes, e.g., by imperfect methylation maintenance in late-replicating domains during mitosis ¹⁸. In contrast, hypermethylation events, e.g., at bivalent promoters, appear to occur in

a more complicated manner. One possible mechanism is the loss of H3K4me3 with age, which directly or indirectly protects bivalent promoters against DNA methylation⁵.

Also, could the authors also address their thoughts on how so few of the DNA methylation changes are in intergenic regions (lines 97-99) and why there is a depletion of interCGI DMRs compared to random?

RESPONSE: As mentioned earlier, RRBS enriches GpC-rich regions of the genome that occur more frequently in the vicinity of genes. Our findings regarding the location of aDMRs go in line with previous reports. It is well-established that DNA hypermethylation predominantly occurs in CpG islands¹⁶. Future studies may be carried out using, e.g., WGBS to cover intergenic regions better. We have clarified this relation in the answer to the previous question and our discussion.

The authors highlight that many of the mid-to-late life DNA methylation transitions are at bivalent promoters- it would be helpful for perhaps a supplemental figure to be included here and could also be used to show if this enrichment was specific to methylation changes at this stage transition or if these bivalent domains were also enriched in other patterns (especially the linear gains). Also the authors should show the OD for SET domain-containing methyltransferase for information about the H3K4me.

RESPONSE: The enrichment of all the identified clusters in the different chromatin states, including bivalent promoters, is presented in Figure 2b. We agree that the labels may not be clear and added a description of abbreviations in the legend. All clusters representing age-associated hypermethylation are generally enriched in bivalent promoters (OR = 10). However, CpGs undergoing a mid-to-late-life transition show an exceptionally high enrichment in these regions (OR = 24). In contrast, the cluster with hypomethylated CpGs is depleted in bivalent regions (OR = 0.1).

Figure 2b shows the enrichment of CpGs in the chromHMM chromatin states predicted by Gorkin et al.¹⁹ using eight histone modification marks (H3K4me3, H3K4me2, H3K27ac, H3K9ac, H3K27me3, H3K4me1, H3K9me3, H3K36me3). The bivalent promoters in the chromHMM are characterized by high signals of H3K4me2, H3K4me3, and H3K37me3. Thus, CpGs undergoing a mid-to-late-life transition are enriched for these three histone marks.

Additionally, we investigated the enrichment of single H3K4me marks. We used the available ChIP-seq data set for H3K4me1, H3K4me2, and H3K4me3 from ENCODE measured in the intestines of postnatal and 2-month-old mice (see the heatmap below). The CpGs undergoing mid-to-late-life transition in C4 and C19 are highly enriched for H3K4me2 and H3K4me3. CpGs undergoing linear hypermethylation changes during aging show weak enrichment for those two marks. Unfortunately, we could not obtain suitable SET domain protein ChIP-seq experiments. We want to address this intriguing question in a follow-up study, e.g., by generating our own data sets.

Minor Concerns:

Figures are difficult to read due to small text- and figure labels in 2B are unclear.

We increased font size in figures: Fig. 2, 5, 6 and explained figure labels in Fig. 2b.

References:

1. Yang, X. et al. Gene body methylation can alter gene expression and is a therapeutic target in cancer. *Cancer Cell* 26, 577–590 (2014).
2. de Mendoza, A. et al. Large-scale manipulation of promoter DNA methylation reveals context-specific transcriptional responses and stability. *Genome Biol.* 23, 163 (2022).
3. Smith, J., Sen, S., Weeks, R. J., Eccles, M. R. & Chatterjee, A. Promoter DNA Hypermethylation and Paradoxical Gene Activation. *Trends Cancer Res.* 6, 392–406 (2020).
4. Tao, Y. et al. Aging-like Spontaneous Epigenetic Silencing Facilitates Wnt Activation, Stemness, and Braf-Induced Tumorigenesis. *Cancer Cell* 35, 315–328.e6 (2019).
5. Kumar, D., Cinghu, S., Oldfield, A. J., Yang, P. & Jothi, R. Decoding the function of bivalent chromatin in development and cancer. *Genome Res.* 31, 2170–2184 (2021).
6. Sun, T. et al. Aging-dependent decrease in the numbers of enteric neurons, interstitial cells of Cajal and expression of connexin43 in various regions of gastrointestinal tract. *Aging* 10, 3851–3865 (2018).
7. Patel, B. A. et al. Impaired colonic motility and reduction in tachykinin signalling in the aged mouse. *Exp. Gerontol.* 53, 24–30 (2014).
8. Patel, B. A. et al. The TNF- α antagonist etanercept reverses age-related decreases in colonic SERT expression and faecal output in mice. *Sci. Rep.* 7, 42754 (2017).
9. Martin, K., Kirkwood, T. B. & Potten, C. S. Age changes in stem cells of murine small intestinal crypts. *Exp. Cell Res.* 241, 316–323 (1998).
10. Franks, L. M. & Payne, J. The influence of age on reproductive capacity in C57BL mice. *J. Reprod. Fertil.* 21, 563–565 (1970).
11. Yu, C. et al. Targeted deletion of a high-affinity GATA-binding site in the GATA-1 promoter leads to selective loss of the eosinophil lineage in vivo. *J. Exp. Med.* 195, 1387–1395 (2002).
12. Ignacio, A. et al. Small intestinal resident eosinophils maintain gut homeostasis following microbial colonization. *Immunity* 55, 1250–1267.e12 (2022).
13. Baumann, A. et al. Microbiota profiling in aging-associated inflammation and liver degeneration. *Int. J. Med. Microbiol.* 311, 151500 (2021).
14. Abu-Jamous, B. & Kelly, S. Clust: automatic extraction of optimal co-expressed gene clusters from gene

- expression data. *Genome Biol.* 19, 172 (2018).
15. Meer, M. V., Podolskiy, D. I., Tyshkovskiy, A. & Gladyshev, V. N. A whole lifespan mouse multi-tissue DNA methylation clock. *Elife* 7, (2018).
 16. Ciccarone, F., Tagliatesta, S., Caiafa, P. & Zampieri, M. DNA methylation dynamics in aging: how far are we from understanding the mechanisms? *Mech. Ageing Dev.* 174, 3–17 (2018).
 17. Meissner, A. et al. Reduced representation bisulfite sequencing for comparative high-resolution DNA methylation analysis. *Nucleic Acids Res.* 33, 5868–5877 (2005).
 18. Zhou, W. et al. DNA methylation loss in late-replicating domains is linked to mitotic cell division. *Nat. Genet.* 50, 591–602 (2018).
 19. Gorkin, D. U. et al. An atlas of dynamic chromatin landscapes in mouse fetal development. *Nature* 583, 744–751 (2020).

REVIEWERS' COMMENTS

Reviewer #1 (Remarks to the Author):

I appreciate the response to my earlier critique and have no further comments and concerns with the revised manuscript

Reviewer #2 (Remarks to the Author):

Thank you to the authors for thoughtfully and robustly addressing the initial concerns raised. The responses have mitigated concerns and more explicitly defined the utility of this model and no additional concerns have been raised by this submission